# Auxiliary Losses for Learning Generalizable Concept-based Models

**Ivaxi Sheth***
CISPA-Helmholtz Center for Information Security
`ivaxi.sheth@cispa.de`

**Samira Ebrahimi Kahou**
École de technologie supérieure, Mila, CIFAR AI Chair
`samira.ebrahimi-kahou@etsmtl.ca`

## Abstract

The increasing use of neural networks in various applications has lead to increasing apprehensions, underscoring the necessity to understand their operations beyond mere final predictions. As a solution to enhance model transparency, Concept Bottleneck Models (CBMs) have gained popularity since their introduction. CBMs essentially limit the latent space of a model to human-understandable high-level concepts. While beneficial, CBMs have been reported to often learn irrelevant concept representations that consecutively damage model performance. To overcome the performance trade-off, we propose cooperative-Concept Bottleneck Model (coop-CBM). The concept representation of our model is particularly meaningful when fine-grained concept labels are absent. Furthermore, we introduce the concept orthogonal loss (COL) to encourage the separation between the concept representations and to reduce the intra-concept distance. This paper presents extensive experiments on real-world datasets for image classification tasks, namely CUB, AwA2, CelebA and TIL. We also study the performance of coop-CBM models under various distributional shift settings. We show that our proposed method achieves higher accuracy in all distributional shift settings even compared to the black-box models with the highest concept accuracy.

## 1 Introduction

The recent advances in deep learning have impacted various domains such as computer vision [21], natural language processing [53], and speech recognition [6]. Recently, the deployment of large models [8, 51] has led to various concerns regarding privacy and safety since machine learning models are often considered black boxes. With the increasing use of such deep learning models in daily human life and their wide deployment, it is essential to understand model behaviors beyond final prediction. Since neural networks are considered opaque decision-makers, inaccurate decisions by models in applications such as medicine [18] or autonomous driving [59] lead to catastrophe for humans. To understand the inner workings of such black-box neural networks, the field of XAI [32] has emerged in recent times. Concept Bottleneck Models (CBMs) [27] are a family of neural networks that enable human interpretable explanations.

Concept-based models introduce a bottleneck layer before the final prediction. This bottleneck layer consists of human-interpretable concept predictions. For example, in the context of images of animals, these concepts could be "mane" in the case of a lion or "black and white stripes" in the case of a zebra.

---

*Work done when the author was at Mila, École de technologie supérieure.

37th Conference on Neural Information Processing Systems (NeurIPS 2023).

Concept bottleneck modelss (CBMs) map input images to interpretable concepts, which in turn are used to predict the label. The intermediary concept prediction allows a human *supervisor* to interpret and understand the concepts influencing the label prediction. In addition to explainability, CBMs offers an interesting paradigm that allows humans to interact with explanations. During inference, a supervisor can query for explanations for a corresponding label, and if it observes an incorrect concept-based explanation then the supervisor can provide feedback.

While CBMs present benefits with models' explainability, Mahinpei et al. [33] have shown that concept representations of CBM may result in information leakage that deteriorates predictive performance. It is also noted that CBM may not lead to semantically explainable concepts [34]. Such bottlenecks may result in ineffective predictions that could prevent the use of CBMs in the wild.

Along with model transparency, another challenge that modern neural networks face is robustness to distributional shifts [32]. Deep learning models fail to generalize in real-world applications where datasets are non-iid [17]. The absence of a comprehensive study examining the behavior of CBMs under distributional shifts is a significant limitation, potentially impeding their application in real-world scenarios.

In this work, we propose **cooperative-CBM** (coop-CBM) model aimed at addressing the performance gap between CBMs and standard black-box models. Coop-CBM uses an auxiliary loss that facilitates the learning of a rich and expressive concept representation for downstream task. To obtain orthogonal and disentangled concept representation, we also propose **concept orthogonal loss** (COL). COL can be applied during training for any concept-based model to improve their concept accuracy. Our main contributions are as follows:

- We proposed a multi-task learning paradigm for Concept Bottleneck Models to introduce inductive bias in concept learning. Our proposed model coop-CBM improves the downstream task accuracy over black box standard models.

- Using the concept orthogonal loss, we introduce orthogonality among concepts in the training of CBMs.

- We perform an extensive evaluation of the generalisation capabilities of CBMs on three different distribution shifts.

- We looked at using human uncertainty as a metric for interventions in CBMs during test-time.

## 2  Related Works

**Concept-based Models**  Early concept-based models that involved the prediction of concepts prior to the classifier were widely used in few-shot learning settings [9, 62]. Other works propose to predict human-specified concepts with statistical modeling [28, 29]. Unsupervised concept learning methods use a concept encoder to extract the concepts and relevance network for final predictions [4, 43]. Although these methods are useful in the absence of pre-defined concepts, they do not enable effective interventions. Concept whitening [11] was introduced as a method to plug an intermediate layer in place of the batch normalization layer of a Convolutional Neural Network (CNN) to assist the model in concept extraction. CBM [27] extends the idea by decomposing the task into two stages: concept prediction through a neural network from inputs, and then target prediction from the concepts. Many works have proposed models built on CBMs to either improve the downstream task accuracy [44, 60, 26] or mitigate the concept leakage [33, 20]. There has been a line of work extending CBMs to real-world applications such as medical imaging [58, 13], autonomous driving [44] and reinforcement learning [14] CBMs require annotated concepts which poses a challenge for their applications to large-scale image datasets. Yuksekgonul et al. [58] propose using concept activation vectors [25] and Oikarinen et al. [38] used multimodal models such as CLIP [40] to annotate concepts for CBMs. Although these either require concept bank or suffer from pretrained model's biases [58].

**Alternative losses**  The training of CBM and its variants typically involves the use of Cross Entropy (CE) loss. Several variants of the CE loss have been explored in the past to improve the discriminative power of learned feature representations of data [19, 46, 56, 39]. Ranasinghe et al. [41], Vorontsov et al. [54] introduce the use of orthogonality in feature space to encourage inter-class separation and intra-class clustering. Our work builds upon [41] by introducing orthogonality constraints in the concept feature space.

# 3 Method

## 3.1 Background

Consider a standard supervised learning setting for a classification task, where models are trained on a dataset $\mathcal{D} = \{x_i, y_i\}_{i=1}^N$ with $N$ data samples. Standard models aim to predict the true distribution $p_{\mathcal{M}}(y|x)$ from an input $x$. Although such a setting has been proven effective on vision benchmarks, users are unaware of the detailed inner workings of the model. Therefore, CBMs introduces intermediate prediction of human-understandable concepts before the model prediction.

In the *supervised concept-based model* setting, the dataset uses additional labeled concept vectors $c_i \in \{0,1\}^a$ where each element indicates the presence of one of $a$ high-level concepts. This allows supervised concept learning in addition to target learning. Following a simplistic causal graph for data generation, $y \rightarrow c \rightarrow x$, CBMs consist of two models. The first model $f_{X \rightarrow C}$ maps the input image $x$ to concepts $c$, while the second model $g_{C \rightarrow Y}$ maps the concepts $c$ to the label $y$.

CBMs can be categorized by their method of training $g_{C \rightarrow Y}$ from the obtained concept representations $f(c|x)$. This could be done in the following manner: *jointly*, where both $f_{X \rightarrow C}$ and $g_{C \rightarrow Y}$ are trained simultaneously end-to-end, *sequentially*, where $f_{X \rightarrow C}$ is trained first, after which $g_{C \rightarrow Y}$ is trained using $p_f(c|x)$ representations, and finally *independently*, where $f_{X \rightarrow C}$ and $g_{C \rightarrow Y}$ are trained individually and then combined.

**Interventions** Interventions are a core motivator of CBMs. The bottleneck model allows for interventions by editing the concept predictions. Since CBMs consider correcting the predicted concepts through interventions during test-time, the corrected concepts are not back-propagated through $f_{X \rightarrow C}$ and $g_{C \rightarrow Y}$. During test-time intervention the predicted concepts can be modified by a supervisor to their ground truth values, leading to "adjusted" concepts prediction. We represent the predicted concepts as $\hat{c} = p_f(c|x)$ and the modified concepts as $\bar{c}$. We consider test-time interventions as an important aspect of explainable models in safety-critical applications. We hypothesize that model-supervisor interaction must lead to the development of a symbiotic relationship between the model and the expert. Here, the expert learns about the potential causation between a concept and its corresponding label, and the model learns true concept values from the expert. We attempt to shine a light on these test-time interventions by simulating realistic scenarios by introducing human uncertainty.

## 3.2 Coop-CBM

While CBMs provide concept explanations behind a prediction, it has been observed that this can come at the expense of lower model accuracy compared to black-box standard models [33]. In this work, we propose a concept-based architecture, coop-CBM to improve the performance of CBMs on downstream classification tasks.

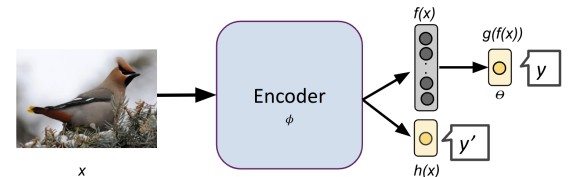

Figure 1: Coop-CBM model that consists of an encoder, concept learner $f$, auxiliary label learner $h$, and task label learning $g$. The encoder transforms input data into a feature representation, which is used by $f$ to predict high-level concepts and $h$ to predict a supplemental auxiliary label, and finally $g$ predicts the final task label conditioned on the concepts only.

**Motivation** The different training paradigms in CBMs introduced by Koh et al. [27] give rise to differences in their concept representations, $p_f(c|x)$. Koh et al. [27] reports that joint CBMs have the highest task accuracy among the different CBM training procedures albeit still lower than standard models. Intuitively, this suggests that joint CBMs, which train both concept predictor and task predictor simultaneously, are able to encode the information about the task label $y$ into concept labels $c$ better than sequential and independent CBMs. In the case of joint CBMs, backpropagation of task loss through the concept predictor aids the overall model in giving more accurate predictions. In this work, we aim to leverage such "soft" information

about the task to improve accuracy. Coop-CBM aims to leverage soft label information in concept predictors to better align the concept predictions to the corresponding label.

**Model** Coop-CBM introduces a multi-task setting before the final prediction. Along with predicting the concepts $c$, we introduce the prediction of task labels in the concept predictor. This allows the model to learn relevant signals and inductive biases of downstream tasks in the concept learning phase. In essence, we now have model $f_{X \to C}$ that predicts concepts $c$ from input $x$ and new model $h_{X \to Y}$ that predicts label $y$ from input $x$. This enables the model to learn relevant knowledge about the task that could be absent in the bottleneck concepts $c$. Although this setting makes the model interpretable, since corresponding concepts to a label can be obtained, one loses the causal $x \to c \to y$ property. Additionally, it does not allow test-time interventions, which is a key application of concept-based models that facilitates human-model interactions. Therefore, to maintain the original properties of CBM, coop-CBM uses a label predictor $g$ which takes the predicted concepts $c$ from $f_{X \to C}$ as input and gives the final label $y$ as output. Hence to avoid confusion, we call the label prediction from $h$ *immediate label*, $y'$, and the final task label, $y$. It must be noted that $f$ and $h$ share all but the last linear layer. Therefore the concept predictor $f$ and $g$ parameters $\theta$, $\phi$ are trained in the following manner:

$$\hat{\theta} = \mathop{\mathbb{E}}_{\mathcal{D}} \left[ \mathop{\arg\max}_{\theta} [\log \ p(c, y'|x; \theta)] \right] = \mathop{\mathbb{E}}_{\mathcal{D}} \left[ \mathop{\arg\max}_{\theta} [\log \ p_f(c|x; \theta) + \log \ p_h(y'|x; \theta)] \right] \tag{1}$$

$$\hat{\phi} = \mathop{\mathbb{E}}_{\mathcal{D}} \left[ \mathop{\arg\max}_{\phi} [\log \ p(y|c; \phi)] \right] \tag{2}$$

Therefore coop-CBM is trained using a linear combination of three different CE losses: $\mathcal{L}_C$ as concept loss, $\mathcal{L}_{y'}$ as immediate label loss and $\mathcal{L}_y$ as task prediction loss.

$$\arg\min[\mathcal{L}_C(f(x), c) + \mathcal{L}_{y'}(h(x), y) + \mathcal{L}_y(g(f(x), y)] \tag{3}$$

In summary, we argue that the introduction of an immediate label introduces the concept predictor to learn meaningful information about the task while still being interpretable. The $h_{X \to Y'}$ model intuitively acts as a regularizer for meaningful concept prediction.

**Mutual information perspective** We hypothesize that by using coop-CBM, the concept predictor acquires better knowledge about $y$. In particular, this can be beneficial when fine-grained concept annotations are not available. We, therefore, suspect that the mutual information (MI) between the input image, concept representations, and the label becomes richer and more expressive as compared to CBM [27]. One way to quantify this is by visualising the MI planes throughout the training, similar to Zarlenga et al. [60].

### 3.3 Concept Orthogonal Loss

Following the current CBM literature, we use cross-entropy loss to train each of the models, $f_{X \to C}$, $h_{X \to Y'}$ and $g_{C \to Y}$ in coop-CBM. In model $f$, each concept is learned via independent and separate classifiers. Given their binary representation, it is intuitive to improve the embedding space of concepts by increasing separability. To do so, we introduce the concept orthogonal loss (COL). By incorporating COL into the training process, we aim to enhance the overall separability of the concept embeddings, leading to improved performance and interpretability of the coop-CBM model.

**Motivation** Due to the variations in training strategies employed in different CBM models, the resulting concept representations can exhibit varying levels of accuracy. The concept accuracy refers to how effectively the learned concept representations align with the ground truth or human-defined concepts. Coop-CBM was concerned with the predictive performance of the task, but here we focus on the concept label accuracy. Higher concept label accuracy signifies improved interpretability. As observed by Koh et al. [27], the concept accuracy of joint CBMs models is lower than other variants because the concepts learned are not completely independent of each other, also called leakage by [33]. Hence, increasing the inter-concept distance and intra-concept clustering throughout the concept vector for the entire dataset can allow the model to learn beyond co-dependent concept representations.

**COL** In addition to CE loss for learning concepts, we introduce novel concept orthogonal loss by conditioning orthogonality constraints on concept feature space. The disadvantage of CE loss is that it does not set a specific distance or separation between different concept representations in the feature space. Consider the CE loss for each concept prediction:

$$\mathcal{L}_{CE}(c, \hat{c}) = \sum_{c_i}^{c_N} -c_i \log(\hat{c}_i) - (1 - c_i)\log(1 - \hat{c}_i) \tag{4}$$

Using this traditional CE loss for each concept, in Equation 4 we are essentially minimizing the difference between the predicted probability distribution and the true probability distribution of the binary concepts. The model, $f_{c_i}$ attempts to learn probability distribution for when a concept is active or inactive respectively. CE does not explicitly enforce separation between concepts.

With concept orthogonal loss (COL), we enforce the separation in the latent representation of the concepts. Our aim with COL is to group similar features together while ensuring that features belonging to different concept classes do not overlap with each other. COL $\mathcal{L}_{\mathcal{COL}}$ enforces inter-concept orthogonality and intra-concept clustering. We define inter-concept separation and intra-concept similarity as $d_1$ and $d_2$ respectively. We enforce orthogonality constrain via cosine similarity. We define the $\mathcal{L}_{COL}$ loss in the shared last layer, $q$ of coop-CBM before concept and auxiliary label predictions. We enforce COL constraints within each batch, $B$.

$$d_1 = \sum_{\substack{i,j \in B, \\ c_i^a = c_j^a \\ a \in A}} \frac{q_i^T q_j}{||q_i|| \; ||q_j||}; \; d_2 = \sum_{\substack{i,j \in B, \\ c_i^a \neq c_j^a \\ a \in A}} \frac{q_i^T q_j}{||q_i|| \; ||q_j||} \tag{5}$$

where $||.||$ denotes the Frobenius norm.

Using the cosine distances, $d_1$ and $d_2$, we simultaneously aim to increase the distance between different latent concept representations and decrease the distance between representations from the same concept. We can introduce a hyperparameter, $\lambda$ to accordingly give weightage to either $d_1$ or $d_2$. The similarity loss $d_1$ between the feature representation of two samples corresponding to the same concept aims to push the $d_1$ towards 1 which means that the feature representations of same concept should be as similar as possible. As for the dissimilarity loss, the goal is to push the loss towards 0, which enforces that the feature representations of different class samples should be as dissimilar as possible. Therefore we consider the absolute value of $d_2$.

$$\mathcal{L}_{COL} = (1 - d_1) + \lambda|d_2| \tag{6}$$

It is important to note that CE loss is applied to each concept binary classification task, which measures the difference between the predicted class probabilities and the true labels. The introduction of COL encourages the network to learn features that are both discriminative and non-redundant among concepts at an intermediary network level. By combining the COL and CE losses, the network is trained to learn discriminative that separate each concept and useful features for classifying when a concept is active. A benefit of COL is it can be universally any concept-based model to encourage orthogonality between different concepts.

In this section, we introduce two auxiliary losses, one to improve the task accuracy using multi-task setting and the other to improve concept representation in latent space, leading to improved concept accuracy. The final loss is a linear combination ($\alpha, \beta, \gamma$ are hyperparameters for weighting in Equation 7) of concept and task losses along with immediate and concept orthogonal losses.

$$\arg\min[\alpha \mathcal{L}_C(f(x), c) + \beta \mathcal{L}_{y'}(h(x), y) + \mathcal{L}_y(g(c), y) + \gamma \mathcal{L}_{COL}(q)] \tag{7}$$

### 3.4 Interventions

Koh et al. [27] demonstrated the potential of CBMs for facilitating human-model interaction and improving task performance during inference. But it can be time-consuming and costly to have domain experts go over each concept, hence some of the recent and concurrent works proposed to use uncertainty as a metric to select interventions.

We propose a lightweight approach that strategizes the supervisor-model interaction. Our method is intuitive and considers three aspects of intervention:

1. Uncertainty of concept prediction - CUS represents the confidence of the model to predict latent concepts, $f_{X \to C}$.

2. Supervisor confidence for concept correction - SCS represents the reliance on the supervisor to intervene and subsequently correct the concepts accurately.

3. Importance of concept for label prediction - CWS denotes the significance of each concept for the subsequent downstream task.

Chauhan et al. [10] propose to optimize interventions over a small validation set using CUS. In comparison, we consider access to the validation set unrealistic. Shin et al. [49], Sheth et al. [48] evaluated interventions more comprehensively and studies the behavior of CBMs during inference by selecting CUS and CWS metrics. We additionally take into account a supervisor's confidence in domain knowledge, SCS and their expertise in correcting the concepts. Concurrent work [12] also looked at human uncertainty for CBMs in depth. Unlike previous works that evaluate test-time interventions on the test splits of respective datasets, we also analyze test-time interventions in OOD setting in the Appendix E.

## 4   Experiments

For our evaluation we consider several image classification benchmark datasets. Our work performs an in-depth empirical analysis of the effectiveness of concept-based models in the presence of different distributional shifts simulating real-world scenarios where data is diverse.

**Baselines**   We consider the models proposed by Koh et al. [27] as our baseline. Additionally, we compare our performance with recent concept-based models that are built on CBM [60, 20]. Due to biases introduced during automatic concept acquisition as mentioned by the authors of Yuksekgonul et al. [58], we consider it to be an unfair baseline to compare generalization properties. They also vary in the number of concepts considered which can also damage the performance and are limited by either the

| Model type | CUB | AwA2 | TIL |
|---|---|---|---|
| Standard *[No concepts]* | 82.3 ±0.2 | 96.2 ±0.1 | 51.1 ±0.9 |
| Independent CBM [27] | 76.0 ±0.4 | 94.9 ±0.3 | 47.4 ±1.0 |
| Sequential CBM [27] | 76.3 ±0.2 | 94.6 ±0.2 | 47.9 ±0.9 |
| Joint CBM [27] | 80.1 ±0.1 | 95.4 ±0.1 | 49.6 ±0.7 |
| CEM [60] | 82.5 ±0.2 | 96.2 ±0.1 | 51.3 ±1.3 |
| CBM-AR [20] | 81.6 ±0.4 | 95.9 ±0.0 | 49.5 ±1.0 |
| Coop-CBM (ours) | 83.6 ±0.3 | 96.6 ±0.1 | 53.4 ±0.8 |
| + COL | **84.1** ±0.2 | **97.0** ±0.1 | **54.2** ±0.9 |

Table 1: Model accuracy on CUB, AwA2 and TIL datasets

presence of concept bank or application (CLIP will fail to generate concepts for TIL dataset), making an unfair comparison.

**Datasets**   We use Caltech-UCSD Birds-200-2011 (CUB) [55] dataset for the task of bird identification. Every dataset image contains 312 binary (eg: beak color, wing color) concepts. We additionally use Animals with Attributes 2 (AwA2) [57] dataset for the task of animal classification. The dataset contains 85 binary concepts. We use all of the subsets of the Tumor-Infiltrating Lymphocytes (TIL) [42] dataset for cancer cell classification.

## 5   Results and Analysis

The primary metric used for the downstream classification task is accuracy. We use the same metric to evaluate the effectiveness of the intervention. We first evaluate the different model performances on the test data split of respective datasets and report the task accuracy $g_{C \to Y}$ for coop-CBM model variants.

**Coop-CBM improves task accuracy**   The evaluation of the performance of a model is based on the final prediction accuracy. In Table 1, we compare the performance of coop-CBM against other baseline models. We first observe that CBMs experienced a significant drop in performance compared

to the standard model that did not use concepts. Our proposed model, coop-CBM with immediate label prediction achieves state of art accuracy and statistically significant results on every dataset. We have observed a significant improvement in the performance of the CUB ($+1.8\%$ increase from the standard model) and TIL($+3.1\%$ increase from the standard model) datasets. This finding is important as it suggests that machine learning models can be designed to overcome the high accuracy vs interpretability tradeoff. Our performance can be further boosted by introducing orthogonality among different concepts. It must be noted that CUB is a fairly densely annotated dataset, which might not always be realistic, hence we also benchmark our model by training on a fraction of concept sets. We also observe a similar trend in results in concept-scarce settings (see Appendix D.2). This suggests that our method is robust to concept selection, which can be beneficial in scenarios where the number of available concepts is limited or expensive to obtain.

**COL improves concept accuracy** Previously in Table 1, we observed that adding concept orthogonal loss to coop-CBM improved its downstream accuracy, in Table 2, we study the impact of adding COL to baseline concept models. Our experiments show that adding COL improves the concept accuracy by a significant margin, especially in joint CBM, CEM, and CBM-AR settings. A known pitfall of CBMs is, *concept leakage* [33] could be potentially prevented by increasing the separation between their concept representations. By maximizing the inner product between the concept embeddings of different concepts, we can ensure that each concept is represented in a separate and distinct direction in the embed-

| Model type | w/o COL | w COL |
|---|---|---|
| Independent CBM [27] | **96.6** $\pm0.0$ | 97.2 $\pm0.1$ |
| Sequential CBM [27] | **96.6** $\pm0.0$ | 97.2 $\pm0.1$ |
| Joint CBM [27] | 93.2 $\pm0.1$ | 96.4 $\pm0.2$ |
| CEM [60] | 94.8 $\pm0.2$ | 97.0 $\pm0.1$ |
| CBM-AR [20] | 94.2 $\pm0.1$ | 96.8 $\pm0.1$ |
| Coop-CBM (ours) | 93.9 $\pm0.2$ | **97.3** $\pm0.2$ |

Table 2: Concept prediction accuracy for each model before and after adding COL for CUB dataset

ding space. This helps in preventing the models from relying on irrelevant concepts. Further to intuitively understand the differences in the concept representation of our model, we compute the histogram for the predicted concept logits. From Figure 3 we see that Coop-CBM+COL minimizes the concept loss better (with help from the auxiliary loss which aids representation learning), which results in clearer separation of logits.

**Clipping concept values to avoid concept leakage** Further, we employ clipping of concept prediction proposed by Mahinpei et al. [33] to further mitigate information leakage in 3 throught two experiments. For the first experiment, we trained the model by clipping the predicted concept values to "hard" labels. Second, we trained the model as we have described earlier in the paper (using soft labels) and evaluated the test set by clipping to "hard" labels. From the above experiments, we conclude that the

| Model | CUB | | | TIL | | |
|---|---|---|---|---|---|---|
| | Std | Exp1 | Exp2 | Std | Exp1 | Exp2 |
| Coop-CBM | 83.6 $\pm0.3$ | 82.1 $\pm0.2$ | 83.0 $\pm0.5$ | 53.4 $\pm0.8$ | 52.6 $\pm0.9$ | 52.8 $\pm1.1$ |
| +COL | 84.1 $\pm0.2$ | 83.2 $\pm0.2$ | 84.0 $\pm0.6$ | 54.2 $\pm0.9$ | 53.5 $\pm0.9$ | 54.1 $\pm0.8$ |

Table 3: Testing for information leakage in our proposed model. Std - standard conditions when joint probabilities are learned to predict the final task, no clipping. Exp1 - During training, we clipped the predicted concept values to "hard" labels. Exp2 - During the evaluation, we clipped the predicted concept values to "hard" labels.

model is able to learn a good representation of the concepts without necessarily leaking information.

**Accounting for Human uncertainty for interventions** As discussed earlier, higher concept accuracy also improves the test-time interventions as seen in Figure 2. While other works used concept weights and uncertainty as metrics to select the interventions, our work introduces a more realistic setting by introducing human uncertainty additionally. The previous works do not account for human error or certainty. Albeit human uncertainty is difficult to quantify since it is often subjective, we use the concept visibility data in the CUB dataset to quantify the confidence score, SCS. The

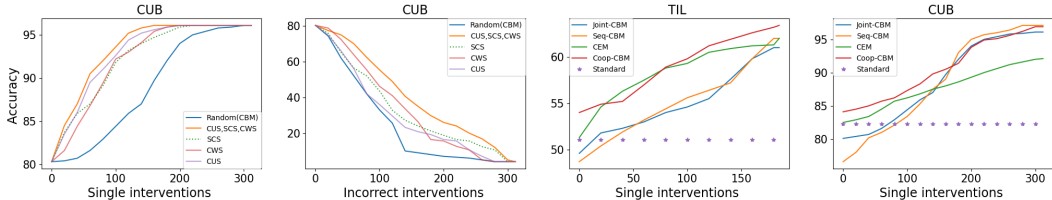

Figure 2: L→R:1) Accuracy vs intervention graph using joint CBM while including supervisor uncertainty. 2)Accuracy vs intervention graph in presence of incorrect interventions by the supervisor using joint CBM. 3) Comparing different model's random interventions on the TIL dataset. 4) Comparing different model's random interventions on the CUB dataset.

annotators rank visibility from $1 - 4$, 1 being 'not visible' and 4 being 'definitely visible'. Figure 2 presents an accuracy curve with an increasing number of interventions on the vanilla CBM model during test-time for proper and fair comparison to [49]. It suggests that considering SCS leads to the most meaningful interventions. On intervention comparison between different concept-based models, we observe that coop-CBM leads to significant interventions that improve downstream performance, especially in TIL dataset where both image descriptive and non-descriptive concepts are present.

In summary, the coop-CBM model and the addition of the concept orthogonal loss help to improve both task and concept accuracy without sacrificing interpretability, which was a common tradeoff in previous methods. This result demonstrates the potential for concept-based models to be more effective in human-AI interactions, especially in domains where expert intervention and interpretability are critical such as healthcare.

## 5.1 Performance under distribution shifts

### 5.1.1 Background spurious correlations

Shortcut-based biases [17] exist in many datasets, where deep learning can easily learn spurious features. In the presence of shortcuts, the model can learn to use spurious features to approximate the true distribution of the labels, as opposed to learning core features. It can be of particular interest to evaluate the performance of concept-based models in the presence of shortcuts. Furthermore, using explainable concepts to facilitate human-model interaction could help reduce the impact of these biases.

The shortcut we consider here is a spurious correlation to the background color in CUB dataset. Loosely following the experimental setup of [5], we correlate the background of a species of bird to its corresponding label. For the dataset, we segment the bird images and add a colored background to all of the images.

| Model | CUB | | CelebA | |
|---|---|---|---|---|
| | In | Out | In | Out |
| Standard | **86.9** ±0.3 | 27.7 ±2.8 | **96.5** ±0.5 | 76.3 ±1.8 |
| Independent CBM [27] | 81.1 ±0.5 | 30.9 ±2.3 | 93.8 ±0.6 | 76.2 ±1.9 |
| Sequential CBM [27] | 81.2 ±0.5 | 30.8 ±2.2 | 93.8 ±0.6 | 76.1 ±1.9 |
| Joint CBM [27] | 83.6 ±0.4 | 32.9 ±3.2 | 94.7 ±0.5 | 80.1 ±1.7 |
| CEM [60] | 84.0 ±0.4 | 34.0 ±2.9 | 95.2 ±0.4 | 81.0 ±1.7 |
| CBM-AR [20] | 82.6 ±0.6 | 33.6 ±2.5 | 95.9 ±0.5 | 79.7 ±2.0 |
| Coop-CBM(ours) | 84.9 ±0.3 | 35.4 ±2.5 | 95.4 ±0.6 | 81.3 ±1.9 |
| + COL | 85.8 ±0.4 | **36.2** ±2.7 | 95.9 ±0.7 | **82.0** ±2.7 |

Table 4: Accuracy of different models under distributional shift - background spurious correlation.

Each class here is correlated to a randomly generated color background with a probability of $80\%$ for the train set. The in-domain test set contains images with similar color background probability as the train set while the correlation in the out-domain test set is reduced to $30\%$. We also consider the hair color-based shortcut induced in Large-scale CelebFaces Attributes (CelebA) dataset where we focus on gender classification between males and females. We aggregate and then construct a modified version of CelebA that learns the shortcut of blonde hair color with women similar to [3].

In Table 4 we report the in-domain and out-domain accuracies of our baseline models and proposed coop-CBM for CUB and CelebA datasets. Our results show the robustness of coop-CBM and COL to background spurious correlations achieving state-of-art results among the concept-based models. We observe an interesting trend for the CUB dataset. The concept-based models may have lower accuracy standard model in the in-domain setting but in the out-domain setting, all of the baseline models including ours have higher accuracy than the standard model. While we do not observe a similar definitive trend in CelebA dataset, it is evident that most of the models, CEM, joint CBM and coop-CBM are superior when the test data contains more images of men with blonde hair. This experiment suggests that the concept-based models are able to generalize better to unseen data outside the training distribution, which can be attributed to their ability to learn more invariant features through the explicit conditioning and learning of concepts. This is a promising result as it indicates that concept-based models, particularly coop-CBM may be more robust to extreme distribution shifts.

In Appendix E, we evaluate the influence of interventions in out-domain settings for the biased CUB dataset. Realistically, the user may fine-tune the model to distributional shift after deployment to improve the predictive performance, but in the circumstances where the labeled shifted dataset is absent, we suspect that interventions could greatly help if they are cheaper to obtain. In such cases, we especially argue that SCS score benefits the intervention selection.

### 5.1.2 Image corruptions

Distributional shift can occur due to various factors such as changes in the data collection process, changes in the environmental conditions under which the data is collected, or even changes in the underlying population that the data represents. The Corruptions dataset [22, 35] is a collection of image corruptions designed to evaluate the robustness of computer vision models. Some of the corruptions are realistic OOD settings such as snow, while others could be less likely in nature such as impulse noise. We introduce 7 such corruptions (Gaussian Noise, Blur, Zoom Blur, Snow, Fog, Brightness and Contrast) onto CUB, AwA2 and TIL datasets. We report detailed results on CUB and average accuracy for AwA and TIL datasets in Table 5.

| Model | CUB | | | | | | | | AwA2 | TIL |
| | 1 | 2 | 3 | 4 | 5 | 6 | 7 | Avg | | |
|---|---|---|---|---|---|---|---|---|---|---|
| Standard | 65.2 ±0.3 | 62.0 ±0.7 | 56.7 ±0.5 | 60.1 ±0.2 | 74.1 ±0.8 | 72.0 ±0.6 | **53.4** ±0.4 | 63.3 | 79.3 | 38.4 |
| Independent CBM [27] | 61.4 ±0.6 | 62.1 ±0.4 | 57.1 ±0.8 | 59.7 ±0.3 | 73.6 ±0.7 | 69.8 ±0.5 | 52.9 ±0.2 | 62.3 | 78.4 | 36.6 |
| Sequential CBM[27] | 60.3 ±0.5 | 61.9 ±0.2 | 56.5 ±0.6 | 58.5 ±0.4 | 72.7 ±0.8 | 71.2 ±0.3 | 52.0 ±0.7 | 61.8 | 78.2 | 36.3 |
| Joint CBM[27] | 63.1 ±0.4 | **64.5** ±0.8 | 57.4 ±0.3 | 60.6 ±0.7 | 73.8 ±0.5 | 72.3 ±0.2 | 51.8 ±0.6 | 63.4 | 79.1 | 37.1 |
| CEM [60] | 66.1 ±0.7 | 61.4 ±0.5 | 57.3 ±0.2 | 61.0 ±0.6 | 74.2 ±0.4 | 71.6 ±0.8 | 53.4 ±0.3 | 63.6 | 79.7 | 38.5 |
| CBM-AR [20] | 64.8 ±0.6 | 61.7 ±0.4 | 57.2 ±0.8 | 59.4 ±0.3 | 73.3 ±0.7 | 70.4 ±0.5 | 52.9 ±0.2 | 62.8 | 79.6 | 36.8 |
| Coop-CBM (ours) | 67.2 ±0.5 | 63.5 ±0.2 | **59.0** ±0.4 | 60.9 ±0.8 | 75.4 ±0.6 | 73.2 ±0.3 | 53.3 ±0.7 | 64.6 | 80.9 | 40.6 |
| + COL | **67.8** ±0.4 | 63.9 ±0.8 | 58.7 ±0.3 | **61.5** ±0.7 | **75.8** ±0.5 | **73.4** ±0.2 | 53.3 ±0.6 | **64.9** | **81.5** | **40.9** |

Table 5: Comparison of concept-based models on image corruptions on CUB, AwA2 and TIL datasets.

Based on our evaluation of distributional shifts in Table 5, we found that incorporating an auxiliary loss in the form of a multi-task setting can help CBMs achieve competitive downstream accuracy overall. We notice that the standard black box models although do perform better in the presence of "contrast" corruption in CUB dataset. Although coop-CBM outperforms other explanation models. In general, it is interesting to note that coop-CBM has better generalization performance in the presence of spurious correlations than in the presence of corruption. This may be because spurious correlations are introduced as a shortcut within the training data, and concept-based models are designed to learn invariant features that are robust to such shortcuts. Regardless, coop-CBM's superior generalization property across different corruptions suggests that the model is able to effectively filter out irrelevant information/noise in the data.

### 5.1.3 Noise concept correlation

In Section 5.1.1 and Section 5.1.2, we conducted experiments concerned with distribution shifts in image space, in this section, we introduce the evaluation of CBMs by simulating distribution shifts in the concept space. To investigate the potential risks of spurious correlations in concept models, we introduced Gaussian noise to the binary concepts. By altering the standard deviation ($\sigma$) of the Gaussian noise, we effectively correlated the shortcut (here noise level) with the image through the concept. To simulate a more realistic setting, instead of adding distinct noise $\sigma$ for each class species, we aggregate random groups of species and add same $\sigma$ to them. In our experiment for the CUB dataset, we add 10 different levels of noise (simulated by $\sigma$) to groups of 20 species labels (200 total classes). For AwA2, we create groups of 10 classes. This approach allowed us to simulate the possibility of introducing unintended correlations between the concepts and the images. By studying the effects of these correlations on the performance of the concept models, we gain insights into the robustness and reliability of the models in handling contaminated concepts. We observe that by introducing a separation between different concepts through COL, our model performs significantly better than the rest of the baselines.

| Model type | CUB | AwA2 |
|---|---|---|
| Independent CBM [27] | 69.7 ±0.6 | 80.1 ±0.4 |
| Sequential CBM [27] | 69.6 ±0.5 | 80.3 ±0.2 |
| Joint CBM [27] | 71.0 ±0.4 | 81.3 ±0.4 |
| CEM [60] | 71.2 ±0.6 | 81.9 ±0.3 |
| CBM-AR [20] | 71.1 ±0.4 | 81.5 ±0.3 |
| Coop-CBM (ours) | 71.9 ±0.6 | 82.5 ±0.2 |
| + COL | **72.7** ±0.4 | **83.2** ±0.2 |

Table 6: Accuracy of different models under distributional shift-noise concept correlation

## 6  Future work and Limitations

In this work, we introduced coop-CBM, a novel concept-driven method to balance AI model interpretability and accuracy. We utilized the Concept Orthogonal Loss (COL) to improve concept learning and applied coop-CBM to various datasets, achieving better generalization, robustness to spurious correlations, and improved accuracy-interpretability trade-offs.

However, our approach has limitations. It relies on labeled concept vectors, which can be challenging in domains with limited annotations and face biases in concept annotation methods. A potential future work could be to extend it to methods that do not assume concept label [58, 38]. Further, we used accuracy as a metric to evaluate concept leakage, in the future, it would be interesting to explore other metrics beyond the accuracy of concept prediction. A future extension of COL could be to evaluate which concepts should be explicitly orthogonalized. We recognize that our model has a few hyperparameters to be optimized. Furthermore, our model assumes that learned concepts align closely with human notions, but this alignment isn't always perfect, affecting comprehensibility. Future research could improve the accuracy of concept-based models by providing meaningful explanations and incorporating additional evaluation metrics. Another potential direction could be to assess the mutual information and therefore establish theoretical grounding to describe the superior performance of coop-CBM.

## 7  Discussion and Conclusion

In this work we proposed two significant contributions to the paradigm of concept-based models. First, we introduced a multi-task model that predicts an intermediary task label along with concept prediction. This is particularly helpful when dense and relevant concept annotation is absent, such as in TIL dataset. Second, we introduced orthogonality constrain in the concept representation space during training via concept orthogonal loss. This loss increases inter-concept separation and decreases intra-concept distance. For both of our proposed methods, we perform extensive experiments on diverse datasets and different distributional shifts. We observe that the bottleneck layer before the final prediction enables concept-based models to exhibit robustness to spurious correlations in the background. Coop-CBM along with COL achieves state-of-art performance for both task accuracy and concept accuracy. Our work indicates that coop-CBM and COL have a strong ability to adapt and generalize well across diverse datasets and real-world scenarios.

## 8 Acknowledgements

We would like to thank Vincent Michalski and the reviewers for engaging in discussions on the earlier version of the paper. The authors would like to thank Google, CIFAR (Canadian Institute for Advanced Research) and NSERC (Natural Sciences and Engineering Research Council of Canada) for supporting and funding the research and Digital Research Alliance of Canada for the compute support.

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

# A  More Related Works

**Exlainability**   Post-hoc explanations aim to provide insight into why a particular prediction or decision was made by the model. These may be in the form of heatmaps, rule sets, or feature importance scores. Global explanations [24] aim to learn the overall generic features in order to explain black-box models by the use of explanators. These explanators are often simple Machine Learning (ML) models. Local explanation methods [1] exhibit explainability by exploring the intrinsic workings of the neural network. This can be executed by propagating layer-wise feature relevance [7]. Selvaraju et al. [47] propose a gradient-based saliency mapping technique that perturbs inputs by injecting noise. Deconstructing the nonlinear model to simpler sub-functions [36] was another proposed method to interpret model's predictions. Nohara et al. [37] propose a difference-to-reference approach to feature importance estimation. Popularly, saliency heatmaps for feature importance visualization have been used [2]. Post hoc explanations although do not address the fundamental issue of model transparency, as they are generated externally to the model and may not reflect the true reasoning of the model's internal mechanisms which can be addressed by concept-based models like ours.

**Orthogonality**   Orthogonality in neural networks has been extensively studied in the literature, with various approaches proposed to enhance model performance and interpretability. X and Y have explored orthogonal regularization techniques, which impose orthogonality constraints on weight matrices or feature representations during training. These methods have been shown to improve generalization and reduce overfitting. Saxe et al. [45] have introduced orthogonal initialization methods, which initialize weight matrices using orthogonal transformations. This initialization strategy has been found to aid training convergence and stabilize the learning dynamics of deep networks. Trockman and Kolter [52] have proposed techniques to enforce orthogonality constraints specifically in convolutional filters of convolutional neural networks (CNNs). By imposing orthogonality on the filters, these methods enhance the representational power and robustness of CNNs. In our work, we introduce orthogonality to concept feature space.

# B  Experiments

The details regarding datasets and hyperparameters are mentioned below. Our codebase is available at `https://github.com/ivaxi0s/coop-cbm` and is built upon from open source repos [27, 41].

## B.1  Dataset Details

**CUB**   The CUB-200-2011 dataset is a collection of 11788 images that are used for fine-grained visual categorization. There are 312 concept attributes that are binarised following the Koh et al. [27] work. While most existing studies use a subset of these concepts, we have chosen to use the entire concept bank in our models and baselines, addressing the fairness issue of subgrouping concepts as highlighted by [49]. The primary task is to classify 200 different species of birds. We also utilize the meta-data of this dataset to obtain human uncertainty.

**TIL**   [42] dataset contains Tumor-Infiltrating Lymphocytes Maps from TCGA HE Whole Slide Pathology Images. The dataset contains tumor maps from the most common cancer tumor types. We use all 13 subsets of the TCGA dataset, therefore constituting 13 cancer types. Although the popular task for such a dataset is necrosis classification, we modify the task to be a classification for different (here 13) tumor types. The advantage of medical images is that their meta-data is readily available from diagnosis. The metadata includes information such as the origin of the tumor, age, gender, and size of tumor cells, which are converted to concepts. We follow [48] for the dataset pre-processing. We wnd up with 185 binary concepts following the pre-processing.

**AwA2**   Animals with Attributes dataset contains over 37,000 images of 50 different animal species, each labeled with 85 distinctive attributes. These attributes can include various characteristics such as color, shape, or behavior, providing a rich source of information for concept bank.

## B.2 Experimental setup

For CUB[55] dataset, we trained using 128 batch size with SGD optimizer with 0.9 momentum and learning rate of $10^{-2}$. The feature extractor was InceptionV3[50] as a concept encoder model.

For AwA2[57] dataset, we trained using 128 batch size with Adam optimizer with 0.9 momentum and learning rate of $5 \times 10^{-3}$. The feature extractor was VIT[15] as a concept encoder model.

For the medical dataset, we create the task of classification divided on the basis of cancer types for TIL dataset[42]. We generate a concept attributes from the meta-data. We use a traditional 70%-10%-20% random split for training, validation, and testing datasets. Additionally, we trained using 64 batch size with SGD optimizer with 0.9 momentum and learning rate of $10^{-2}$. The feature extractor was InceptionV3[50] as a concept encoder model.

For m-CelebA[31] dataset, we train using 64 batch size with Adam optimizer with 0.9 momentum and learning rate of $5 \times 10^{-3}$ for 500 epochs. The feature extractor was InceptionV3[50] as a concept encoder model.

Across all of the models for tasks, we use weight decay of factor of $5 \times 10^{-5}$ and scale the learning rate by a factor of 0.1 if no improvement has been seen in validation loss for the last 15 epochs during training. We also train using an early stopping mechanism i.e. if the validation loss does not improve for 200 epochs, we stop training.

In this paper, we introduced two auxiliary losses, one to improve the task accuracy using multi-task settings and the other to improve concept representation in latent space, leading to improved concept accuracy. The final loss is a linear combination ($\alpha, \beta, \gamma$ are hyperparameters for weighting in Equation 7) of concept and task losses along with immediate and concept orthogonal losses.

$$\arg\min[\alpha \mathcal{L}_C(f(x), c) + \beta \mathcal{L}_{y'}(h(x), y) + \mathcal{L}_y(g(c), y) + \gamma \mathcal{L}_{COL}(q)] \tag{8}$$

For the hyper-parameters of Equations 7, we use $\alpha$ and $\beta$ values to 0.01 for all of the experiments.

| Model | Task | Concept |
|---|---|---|
| $\alpha = \beta = \gamma = 0.01$ | 84.1 | 97.3 |
| $\alpha = \beta = \gamma = 1.0$ | 83.0 | 96.8 |
| $\alpha = \beta = 0.1, \gamma = 0.01$ | 83.4 | 97.4 |
| $\alpha = \beta = 0.01, \gamma = 0.1$ | 83.2 | 97.4 |
| $\alpha = \beta = 0.1, \gamma = 0.1$ | 83.4 | 97.5 |
| $\alpha = 0.1, \beta = \gamma = 0.01$ | 84.0 | 97.3 |
| $\alpha = 0.01, \beta = 0.1, \gamma = 0.01$ | 84.2 | 97.0 |

Table 7: Different weightage - coop-cbm with COL on CUB dataset

**CEM hyperparameters** We would like to point out that we used the same concept $\alpha$ weightage hyperparameter for each of the model. In literature, for all [27, 60, 20] of the methods used the same concept weightage and we follow the same convention. Looking at our selected hyperparameter, the most divergent value is for CEM [60]. The original CEM paper selected the $\alpha = 5$. Since we could not find an ablation study around this hyperparameter in the original paper, we continued to use the same value as rest of the models. We acknowledge that our results might be skewed due to this reason.

## B.3 Resources used

Our codebase was built upon the open codebase of [27]. We trained on Linux-based clusters mainly on V100 GPUs and partially on A100 GPU. The following table includes compute timing for each epoch for every baseline.

| Model type | CUB | TIL | AwA2 |
|---|---|---|---|
| Standard | 57s | 33s | 272s |
| CBM | 68s | 39s | 286s |
| CEM | 78s | 46s | 297s |
| CBM-AR | 87s | 50s | 313s |
| Coop-CBM (ours) | 61s | 41s | 289s |

Table 8: Compute timings of baseline models on different datasets on a V100 GPU.

## C  Further experiments on COL

### C.1  Disentanglement

Disentangled features allow for a more intuitive understanding of the underlying factors that influence the concepts. Several research efforts have explored the benefits and applications of disentangled representations [23]. One of the notable characteristics of Concept Orthogonal Loss (COL) is its ability to induce disentanglement in the concept space. By incorporating COL into the training process, the model learns to separate and represent concepts in a more distinct and independent manner. This disentanglement is achieved by enforcing an orthogonal relationship among different concept representations. To evaluate disentanglement, we use the Oracle Impurity Score as proposed by Zarlenga et al. [61]. The metric essentially aims to detect for impurities in soft representations of concepts. We use this metric to compare against Joint-CBM. The OIS score for Joint-CBM on the CUB dataset was $0.19$ while on coop-CBM with COL $0.14$ which shows better disentanglement in concept learning for coop-CBM + COL model showing the benefit of using COL on top of any model ad-hoc.

COL encourages the model to assign orthogonal directions to different concepts, thereby reducing the overlap and correlation between them. As a result, each concept becomes more independent and captures a specific aspect or attribute of the input data. This disentanglement in the concept space enables better interpretability and facilitates a clearer understanding of how different concepts contribute to the model's decision-making process.

Through disentanglement, COL enhances the separability and discriminative power of the learned concept representations. It allows the model to focus on relevant and informative aspects of the data while minimizing the influence of irrelevant or redundant features. This disentanglement not only improves the interpretability of the model but also contributes to its overall performance by reducing concept interference and enhancing the model's ability to generalize to new and unseen data.

By promoting disentanglement in the concept space, COL provides a valuable tool for understanding and analyzing the inner workings of concept-based models. It opens up opportunities for further research and exploration into how disentangled concept representations can be leveraged for various tasks, including transfer learning, domain adaptation, and model debugging.

### C.2  When is concept orthogonality relevant

One obvious question and could be an interesting future work could be to devise what explicit concepts must be orthogonal. Our work assumes that every concept must be orthogonal but potentially there could be an application where it could be beneficial to include partial orthogonality. Devising an optimal point for when to use COL could be great future work building on our work. We attempt to provide justification using empirical analysis and some intuition in this section.

**Duplicating concepts**  We consider a scenario where input concepts are intentionally duplicated to create a high degree of concept correlation. In our experiment from Table 9, we duplicated $10\%$, $25\%$, $50\%$ and $100\%$ of concepts and added them to the original concept bank. This is a worst-case representation of "similar concepts". From the table, we see that the duplication of concepts does not impact the concept or the task accuracy. Additionally, this experiment contributes to the broader understanding of how COL performs in various scenarios. We observe that the performance is not significantly impacted for concept duplication if we add COL.

|  | 10% | 25% | 50% | 100% |
|---|---|---|---|---|
| Task accuracy CUB | 83.9 ±0.4 | 83.7 ±0.7 | 83.2 ±0.5 | 83.4 ±0.2 |
| Task accuracy TIL | 54.0 ±0.7 | 54.1 ±1.1 | 53.8 ±1.2 | 53.8 ±0.1.1 |

Table 9: Evaluating the robustness of COL loss in the presence of concept correlation. We randomly duplicate a percentage of the concept bank and evaluated our model Coop-CBM+COL on the CUB and TIL datasets.

**Histogram of concept logits for joint-CBM and coop-CBM**   To gain further insight into the effect of COL, we computed histograms of the activations of the penultimate layer (to which the loss is applied) and saw that the histogram of CBM+COL is very sparse (with a large peak at 0 and a much smaller one at 1) in contrast to the vanilla CBM.

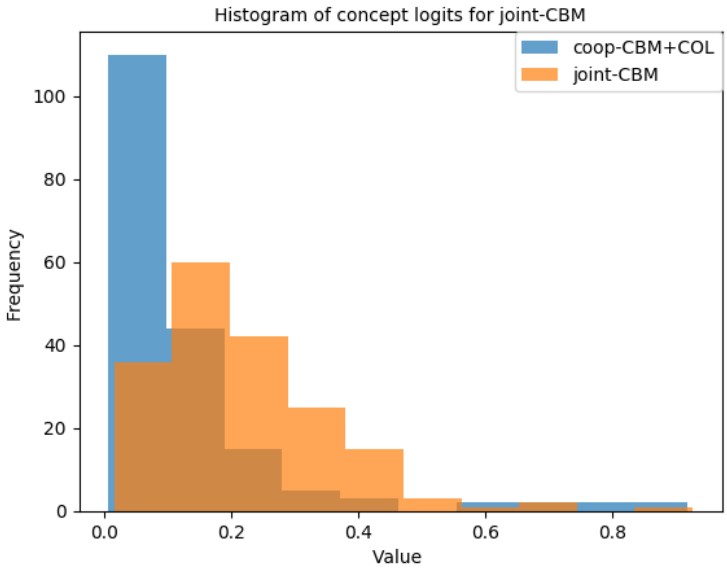

Figure 3: Concept logits histogram comparison on TIL dataset

**Intuition**   The dissimilarity loss $d_2$ encourages independent concept prediction. This is particularly important to reduce leakage in CBMs and improve the robustness of concept explanations. For the connection between entanglement of concept predictions and leakage, we refer the reader to the introduction section of Havasi et al. [20]. Motivated by this insight, we introduce $d_2$, to encourage disentanglement of the penultimate layer of features for concept prediction. We agree that this may result in an overcomplete representation, induced by the opposing forces of $d_1$ and $d_2$ for samples with partially overlapping concepts, and multiple feature groups may contribute to the same concept. However, it might be difficult to achieve disentanglement otherwise, and as our experiments show, COL improves the concept representation and accuracy, including in the out-of-domain settings. To gain further insight into the effect of COL, we computed histograms of the activations of the penultimate layer (Figure 3) and see that the histogram of CBM+COL is very sparse (with a large peak at 0 and a much smaller one at 1) in contrast to the vanilla CBM. This suggests that COL may encourage learning of an overcomplete sparse feature space, the elements of which encode various combinations of concepts, and the last layer of $f$ learns to introduce invariance in the prediction of each concept $c_i$ with respect to specific combinations with other concepts $c_j$ by linear combination. Additionally, it must be noted that this regularization is only applied to the penultimate layer before concept prediction which means low-level features are still free to share weights. Essentially, we believe COL encourages learning a sparse over-complete dictionary of features with concepts still partially entangled in different combinations. The concept prediction layer in $f$ then learns a linear combination of these specialized features to introduce invariance

with respect to the specific combinations. In fact, when studying histograms of activations of the penultimate layer, we observe that activations with COL are indeed very sparse in contrast to CBMs without COL. Given a sparse dictionary of combinations of concepts as induced by COL, the task of disentanglement of concept prediction would *ideally* reduce to a linear combination of dictionary elements. This is the intuition behind sparse coding (see [30]). In our case, the sparsity is induced indirectly as a result of the orthogonality-based loss formulation. Our experiments show that this approach significantly facilitates the overall optimization of CBMs, improving concept accuracy and downstream performance.

### C.3 Effect of lambda on COL

We experimented with different loss weights for $\lambda$ in our experiments and the model+COL seemed to be fairly robust with different values of $\lambda$. We have put those results in Table 5 of the rebuttal PDF on CUB and TIL datasets. While a fine-tuned value of $\lambda$ might show good performance, we observe that regardless, the model is still able to beat the performances of other baselines. We observed that 0.05 set as a good tradeoff between performance and uncertainty across datasets.

| Dataset | $\lambda$=0.05 | $\lambda$=0.1 | $\lambda$=0.5 | $\lambda$=1.0 | $\lambda$=10.0 |
|---|---|---|---|---|---|
| CUB | 84.1 ±0.2 | 84.1 ±0.4 | 83.8 ±0.3 | 84.0 ±0.5 | 83.6 ±0.3 |
| TIL | 54.2 ±0.9 | 54.0 ±0.8 | 54.3 ±0.6 | 53.6 ±0.8 | 54.1 ±1.0 |

Table 10: Effect of $\lambda$ on the coop-CBM+COL model. We observe that the orthogonal loss is fairly robust to hyperparameter selection.

## D    Further experiments

### D.1    Comparison against models with automated concept acquisition

[38] and [58] used a pre-trained model - CLIP which was trained on a massive corpus of data to obtain concepts. This can potentially introduce inherent biases from pretraining into the concepts. This was also brought up in the Limitations and Conclusion section of [58]. Furthermore, the dissimilarity in the concepts employed in these works adds complexity to establish a fair and meaningful comparison. Moreover, we wish to emphasize that neither of these works directly compare with CBM variants in their main paper, except for [58] which appears in Appendix C. Also it is not possible to compare with realistic medical datasets as CLIP fails to generate meaningful concepts. Regardless, we have compared both the methods with our method on CUB+OOD datasets and our model outperforms the accuracy of [38] and [58], which is lower than the standard model. We believe it will be interesting future to include our model methodology and COL in the respective models.

| Model type | CUB | OOD-CUB | corr-CUB |
|---|---|---|---|
| Standard | 82.3 | 27.7 | 63.3 |
| PCBM | 78.4 | 33.4 | 62.7 |
| PCBM-h | 80.9 | 32.1 | 62.9 |
| LF-CBM | 81.0 | 33.8 | 63.9 |
| Coop-CBM | 83.6 | 35.4 | 64.6 |
| + COL | **84.1** | **36.2** | **64.9** |

Table 11: Comparison of Posthoc CBM and Label-Free CBM with our proposed methodology. OOD-CBM refers to Exp 5.1 relating to spurious correlation generalization. Coor-CBM refers to Exp 5.2 relating to image corruption. Unfortunately, Exp 5.3 could not be conducted due to the different concept bank.

## D.2 Coop-CBM and COL in the presence of sparse concept labels

Concept labeling could be a labor-intensive task and hence it is important to understand the most optimal point of operation. We randomly select a subset of concepts and train baselines on the subset. Due to concept and task prediction at the same level, we observe coop-CBM provides inductive bias for the downstream task.

| Model | 10% | 25% | 50% | 100% |
|---|---|---|---|---|
| Standard | 82.3 ±0.4 | 82.3 ±0.5 | 82.3 ±0.3 | 82.3 ±0.2 |
| Independent CBM [27] | 33.6 ±0.5 | 62.0 ±0.3 | 74.7 ±0.1 | 76.0 ±0.2 |
| Sequential CBM[27] | 45.5 ±0.3 | 62.0 ±0.4 | 74.3 ±0.2 | 76.3 ±0.2 |
| Joint CBM[27] | 75.2 ±0.4 | 78.5 ±0.5 | 79.6 ±0.3 | 80.1 ±0.1 |
| CEM [60] | 76.1 ±0.5 | 79.3 ±0.2 | 79.5 ±0.1 | 82.5 ±0.2 |
| CBM-AR [20] | 73.4 ±0.4 | 77.8 ±0.4 | 79.7 ±0.3 | 81.6 ±0.4 |
| Coop-CBM (ours) | 81.4 ±0.3 | 82.9 ±0.5 | 83.7 ±0.4 | 83.6 ±0.3 |
| + COL | 82.4 ±0.4 | 83.2 ±0.3 | 83.9 ±0.5 | 84.1 ±0.2 |

Table 12: Task accuracy CUB with sparse concept annotations (% - fraction of concepts)

| Model | 10% | 25% | 50% | 100% |
|---|---|---|---|---|
| Independent CBM [27] | 94.2 ±0.2 | 95.3 ±0.1 | 96.7 ±0.3 | 96.6 ±0.0 |
| Sequential CBM[27] | 94.2 ±0.1 | 95.3 ±0.3 | 96.7 ±0.2 | 96.6 ±0.1 |
| Joint CBM[27] | 90.7 ±0.3 | 92.0 ±0.2 | 93.5 ±0.1 | 93.2 ±0.1 |
| CEM [60] | 93.6 ±0.2 | 93.9 ±0.1 | 94.1 ±0.3 | 94.8 ±0.2 |
| CBM-AR [20] | 93.1 ±0.1 | 93.7 ±0.3 | 94.0 ±0.2 | 94.2 ±0.1 |
| Coop-CBM (ours) | 92.5 ±0.3 | 92.9 ±0.2 | 93.6 ±0.1 | 93.9 ±0.2 |
| + COL | 96.6 ±0.2 | 96.8 ±0.1 | 97.1 ±0.3 | 97.3 ±0.2 |

Table 13: Concept accuracy CUB with sparse concept annotations (% - fraction of concepts)

| Model | 10% | 25% | 50% | 100% |
|---|---|---|---|---|
| Standard | 51.1 ±1.2 | 51.1 ±0.9 | 51.1 ±0.7 | 51.1 ±0.9 |
| Independent CBM [27] | 15.4 ±1.3 | 25.6 ±0.8 | 43.2 ±1.1 | 47.4 ±1.0 |
| Sequential CBM[27] | 19.8 ±0.7 | 27.9 ±1.4 | 43.9 ±1.0 | 47.9 ±0.9 |
| Joint CBM[27] | 43.5 ±1.5 | 44.9 ±0.6 | 46.1 ±1.2 | 47.6 ±0.7 |
| CEM [60] | 46.1 ±0.9 | 47.9 ±1.3 | 49.2 ±0.7 | 51.3 ±1.3 |
| CBM-AR [20] | 46.1 ±1.1 | 48.8 ±0.8 | 49.2 ±1.5 | 49.5 ±1.0 |
| Coop-CBM (ours) | 50.3 ±1.4 | 51.7 ±0.7 | 52.8 ±1.2 | 53.4 ±0.8 |
| + COL | 51.0 ±1.0 | 52.2 ±1.5 | 53.5 ±0.6 | 54.2 ±0.9 |

Table 14: Task accuracy TIL with sparse concept annotations (% - fraction of concepts)

| Model | 10% | 25% | 50% | 100% |
|---|---|---|---|---|
| Independent CBM [27] | 92.3 ±0.5 | 94.5 ±0.3 | 95.7 ±0.7 | 96.4 ±0.2 |
| Sequential CBM[27] | 92.3 ±0.6 | 94.5 ±0.4 | 95.7 ±0.8 | 96.4 ±0.3 |
| Joint CBM[27] | 91.7 ±0.7 | 92.3 ±0.5 | 93.2 ±0.2 | 93.9 ±0.6 |
| CEM [60] | 93.2 ±0.8 | 93.8 ±0.6 | 94.2 ±0.3 | 94.4 ±0.7 |
| CBM-AR [20] | 93.6 ±0.2 | 93.8 ±0.8 | 94.2 ±0.5 | 94.2 ±0.6 |
| Coop-CBM (ours) | 93.2 ±0.3 | 93.8 ±0.7 | 93.9 ±0.4 | 94.2 ±0.8 |
| + COL | 96.4 ±0.4 | 96.6 ±0.2 | 97.0 ±0.6 | 97.1 ±0.5 |

Table 15: Concept accuracy TIL with sparse concept annotations (% - fraction of concepts)

| Model | 10% | 25% | 50% | 100% |
|---|---|---|---|---|
| Standard | 96.2 ±0.2 | 96.2 ±0.1 | 96.2 ±0.6 | 96.2 ±0.1 |
| Independent CBM [27] | 42.4 ±0.4 | 67.6 ±0.2 | 89.4 ±0.1 | 94.9 ±0.3 |
| Sequential CBM[27] | 52.6 ±0.3 | 71.5 ±0.6 | 91.7 ±0.2 | 94.6 ±0.2 |
| Joint CBM[27] | 89.2 ±0.1 | 91.8 ±0.4 | 94.0 ±0.3 | 95.4 ±0.1 |
| CEM [60] | 89.9 ±0.6 | 93.1 ±0.3 | 95.5 ±0.1 | 96.2 ±0.1 |
| CBM-AR [20] | 91.0 ±0.2 | 92.6 ±0.1 | 93.8 ±0.4 | 95.9 ±0.0 |
| Coop-CBM (ours) | 92.6 ±0.3 | 94.2 ±0.1 | 96.1 ±0.2 | 96.6 ±0.1 |
| + COL | 92.9 ±0.1 | 95.4 ±0.1 | 96.5 ±0.3 | 97.0 ±0.6 |

Table 16: Task accuracy AwA2 with sparse concept annotations (% - fraction of concepts)

| Model | 10% | 25% | 50% | 100% |
|---|---|---|---|---|
| Independent CBM [27] | 94.7 ±0.2 | 95.8 ±0.1 | 97.3 ±0.4 | 97.7 ±0.3 |
| Sequential CBM[27] | 94.7 ±0.3 | 95.8 ±0.2 | 97.3 ±0.1 | 97.7 ±0.4 |
| Joint CBM[27] | 93.1 ±0.4 | 94.2 ±0.3 | 94.8 ±0.2 | 95.2 ±0.1 |
| CEM [60] | 92.6 ±0.1 | 93.2 ±0.4 | 94.8 ±0.3 | 95.6 ±0.2 |
| CBM-AR [20] | 93.5 ±0.2 | 93.7 ±0.1 | 95.0 ±0.4 | 95.4 ±0.3 |
| Coop-CBM (ours) | 93.5 ±0.3 | 94.6 ±0.2 | 95.2 ±0.1 | 95.7 ±0.4 |
| + COL | 96.6 ±0.4 | 96.8 ±0.3 | 97.1 ±0.2 | 98.4 ±0.1 |

Table 17: Concept accuracy AwA2 with sparse concept annotations (% - fraction of concepts)

### D.3 Detailed results with image corruptions

| Model | TIL | | | | | | | Avg |
|---|---|---|---|---|---|---|---|---|
| | 1 | 2 | 3 | 4 | 5 | 6 | 7 | |
| Standard | 35.5 ±1.2 | 34.7 ±1.7 | 39.3 ±1.4 | 35.7 ±1.1 | 41.6 ±1.6 | 39.9 ±1.3 | 43.9 ±1.0 | 38.4 |
| Independent CBM [27] | 32.8 ±1.5 | 31.9 ±1.0 | 39.5 ±1.7 | 34.3 ±1.2 | 40.8 ±1.1 | 32.9 ±1.6 | 43 ±1.3 | 36.6 |
| Sequential CBM[27] | 33.0 ±1.3 | 31.6 ±1.4 | 39.9 ±1.1 | 34.6 ±1.6 | 40.2 ±1.5 | 32.4 ±1.0 | 42.9 ±1.7 | 36.3 |
| Joint CBM[27] | 33.3 ±1.4 | 32.3 ±1.3 | 41.6 ±1.2 | 38.2 ±1.7 | 41.2 ±1.0 | 33.3 ±1.5 | 39.8 ±1.1 | 37.1 |
| CEM [60] | 35.2 ±1.1 | 34.8 ±1.6 | 40.0 ±1.5 | 37.3 ±1.0 | 42.1 ±1.7 | 33.6 ±1.2 | 46.5 ±1.3 | 38.5 |
| CBM-AR [20] | 35.4 ±1.7 | 35.1 ±1.2 | 39.7 ±1.3 | 38.4 ±1.4 | 41.6 ±1.1 | 42.8 ±1.6 | 25.6 ±1.5 | 36.8 |
| Coop-CBM (ours) | 36.7 ±1.0 | 36.2 ±1.7 | 41.4 ±1.2 | 37.6 ±1.3 | 43.3 ±1.4 | 39.9 ±1.1 | 49.1 ±1.6 | 40.6 |
| + COL | 37.2 ±1.6 | 36.5 ±1.5 | 41.5 ±1.0 | 38.3 ±1.7 | 43.0 ±1.2 | 44.4 ±0.9 | 45.4 ±1.0 | 40.9 |

Table 18: Comparison of concept-based models on image corruptions on TIL datasets

## E  Further about interventions

**Concept Uncertainty Score**

A significant part of the intervention selector constitutes Concept Uncertainty Score (CUS) which denotes the model uncertainty for the prediction of concepts. We calculate epistemic uncertainty that arises due to model parameters and lack of training samples. Realistically, labeled medical data is often scarce, encouraging the application of epistemic uncertainty quantification of the predicted concepts. We use Monte-Carlo dropout [16] to model epistemic uncertainty, with a random dropout rate of 0.2. We apply the dropout before the prediction of concepts. For an image $x_i$ predicting concepts $c_i...c_K$ where $K$ is the total number of concepts, we evaluate $T$ softmax probabilities $\{p_t\}_{t=1}^{T}$ for each concept prediction. We measure the uncertainty for each concept which we refer to as $\mathcal{H}(\cdot)$. We compute the entropy-based uncertainty for each concept as the measure of the expectation of the information inherited in the possible outcomes of a random variable. Using the uncertainty metric, we calculate the overall uncertainty concept vector $\mathcal{H}$.

$$\mathcal{H}(\cdot) = -\frac{1}{N} \sum_{i=1}^{N} \left( \frac{1}{T} \sum_{t=1}^{T} p_g(c|x_i) \log \frac{1}{T} \sum_{t=1}^{T} p_g(c|x_i) \right) \tag{9}$$

**Concept Weightage Score**

The second part of the intervention selector score is signified by Concept Weightage Score (CWS), which accounts for the importance of a concept in the final downstream prediction task. Using CWS, the intervention selector is able to prioritize the concept for intervention that is deemed to change the final prediction significantly. We define the weightage score as $\beta$. The $f(y|c)$ is a linear one-layer Multi-Layer Perceptron (MLP) which helps in defining $\beta$.

$$\beta_i = c_i \sum_{j=1}^{N} |w_{ij}| \tag{10}$$

**Supervisor Confidence Score**

Finally, in the intervention selector, we consider the reliability of the annotator via Supervisor Confidence Score (SCS). For example, while a histopathologist can identify diseases across human tissues and organs, they often have more specialized and nuanced areas of focus. It is therefore

beneficial for the model to request additional information from histopathologists from their expert knowledge. This in practice prevents ambiguous or inaccurate concept correction. The SCS is a variable across each annotator and is represented by $\gamma$.

Finally, bringing the three desiderata of concept selection for intervention, the final intervention selector score is a linear combination of all of the scores.

$$AISelect = k_1 * \left( \frac{1}{N} \sum_{i=1}^{N} \left( \frac{1}{T} \sum_{t=1}^{T} p_g(c|x_i) \log \frac{1}{T} \sum_{t=1}^{T} p_g(c|x_i) \right) \right) + k_2 * \sum_{i=1}^{K} c_i \sum_{j=1}^{N} |w_{ij}| + k_3 * \gamma$$

(11)

where $k_1, k_2, k_3$ are hyperparameters for importance weightage on each of the scores.

Intervention allows the model to query the most significant concepts. In our case, it is hypothesized that the most uncertain concepts will build a symbiotic relationship between the human and the model. The supervisor decides the threshold $\mathcal{I}_{th}$ to correct the concept prediction. Realistically, in medical scenarios, due to the professional's limited availability, we would like to optimize the number of concepts to intervene. Setting a lower threshold is a trade-off decided by the user. In contrast to other works that perform group interventions [27] by intervening on a group of similar concepts, we perform single interventions. Group interventions require clustering of concepts on the basis of their similarity[2], which is not realistic as such information is not always available. Therefore, single interventions are performed to minimize the dependence on human priors.

---

**Algorithm 1** Intervention selector Pseudocode

---

$X \leftarrow$ input image
$c_1...c_n \leftarrow$ n intermediary concepts
$Y \leftarrow$ label
$g \leftarrow$ Image to concept prediction model
$f \leftarrow$ Concept to label prediction model
$c_1...c_n = g(X)$
**for** $i = 1, 2, \ldots, n$ **do**
$\quad \mathcal{H}_i = CUS(c_i)$
$\quad \beta_i = CWS(c_i)$
$\quad \gamma_i = SCS(c_i)$
$\quad \hat{c}_i = k_1 * \mathcal{H}_i + k_2 * \beta_i + k_3 * \gamma_i$
**end for**
$\hat{c}_1...\hat{c}_{thr}...\hat{c}_n \leftarrow$ threshold
$\bar{c}_1...\bar{c}_{thr} \leftarrow$ intervene on threshold valued
$Y' = f(\bar{c}_1...\bar{c}_n)$

---

# F   Example of OOD data

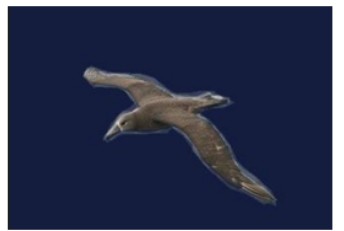 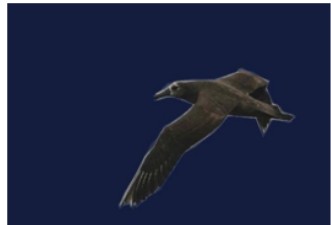 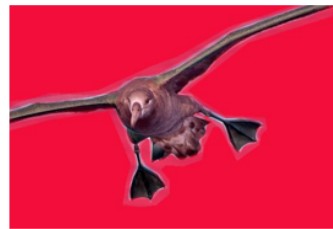

Figure 6: L: example from trainset, M: example from in-domain testset, R: example from out-domain testset from spurious background correlation CUB synthetic dataset, class - Black Footed Albatross

---

[2]This similarity metric is pre-defined by humans

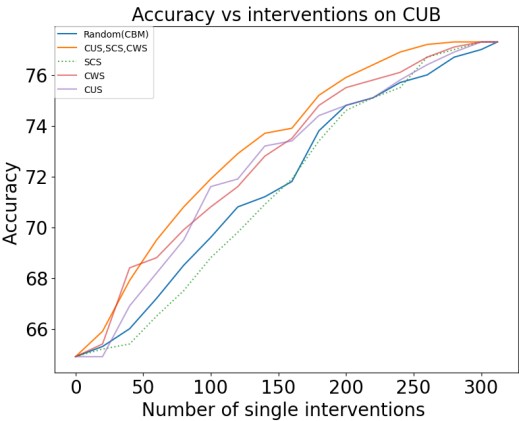

Figure 4: Interventions on Joint CBM in the presence of image corruptions on CUB dataset

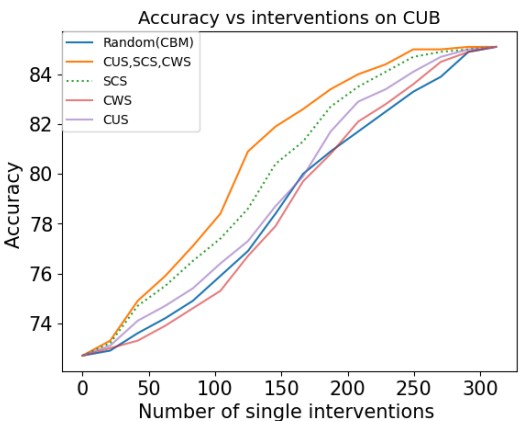

Figure 5: Interventions on Joint CBM in the presence of image corruptions on CUB dataset

# G  Limitations and Future Work

In this work, we proposed a novel concept-based approach, coop-CBM, to enhance the interpretability and accuracy trade-off in AI models. We introduced the Concept Orthogonal Loss (COL) to improve concept learning and employed Coop-CBM on various datasets and evaluation scenarios. Our results demonstrated superior generalization, robustness to spurious correlations, and advancements in the accuracy-interpretability trade-off.

While our proposed approach has shown promising results and made significant contributions, it is important to recognize certain limitations. Firstly, the reliance on labeled concept vectors poses challenges in domains where concept annotations are limited or costly to acquire. Furthermore, current concept annotation methods suffer from biases and a lack of domain knowledge, indicating the need for further improvements in this area. Our work does not address unsupervised concept acquisition methods, but instead focuses on a model architecture that can be applied regardless of the concept acquisition approach.

While our concept-based approach offers interpretability, it assumes that the learned concepts align closely with human-understandable notions. However, there is a possibility that the learned concepts may not always perfectly align with the intended interpretations, which can pose challenges in terms of comprehensibility and explainability. Future research could delve deeper into understanding explanations and their alignment with human understanding, thereby exploring ways to improve the fidelity of concept-based models in providing accurate and meaningful explanations. Future research could aim to incorporate additional evaluation metrics that assess the transparency, fairness, and robustness aspects of concept-based models.

Overall, while our approach shows promising results and addresses important concerns in the field of explainable AI, it is important to be aware of these limitations and continue advancing research to overcome them and further enhance the applicability and reliability of concept-based models.

