# OpenReview forum: "Auxiliary Losses for Learning Generalizable Concept-based Models"
_NeurIPS.cc/2023/Conference — NeurIPS 2023 poster_

### Official Review · Reviewer_VWed · 2023-06-18

**Soundness:** 3 good
**Presentation:** 3 good
**Contribution:** 3 good
**Rating:** 6
**Confidence:** 3

**Summary:**

The paper proposes multiple improvements to current paradigm about learning, application and evaluation of concept bottleneck models (CBMs) for by-design interpretable classification. They propose to improve supervised learning of concepts in CBMs through a concept-orthogonality loss (COL) that encourages samples from different classes to have orthogonal feature representations used for concept prediction and samples with same concepts to have similar representation. Addtionally, to improve classification performance of the model, they propose to predict class labels additionally from this representation. In terms of usage of CBMs, the authors propose to incorporate supervisors confidence when making decisions about intervention. In terms of evauation they propose to also evaluate baselines under distribution shifts.

**Strengths:**

Strengths:
(a) The paper is well written in large parts. The organization is fluid and comfortable to follow. The motivations for the proposals are also generally clear.

(b) The experiments are solid and extensive with multiple different datasets, baselines and experimental settings.

(c) The paper contains multiple novel contributions in terms of learning, usage or evaluation of CBMs.

The quality and clarity is high in most parts. Considering all the contributions as a whole, the paper is reasonably original and moderately significant in my opinion.

**Weaknesses:**

Weaknesses:
(a) Each individual contribution is limited in either novelty or impact. For instance the concept-orthogonality loss (COL) is the most novel contribution but I wouldn't consider the improvement in concept/model accuracy (Tables 1-4) highly significant. On the other extreme while I'd consider evaluation of all baselines under distribution shifts useful for community, I won't regard it as highly original. This weakness sets an upper bound in terms of strength of this paper for me. That being said, considering all contributions together, this won't be a determining factor for me in terms of recommending acceptance or rejection.

(b) The experiments studying impact of loss weights are lacking in certain ways. More details  about this in section for questions.

(c) Statements and reasoning around concept orthogonality-loss (COL) is sometimes confusing and potentially misleading to me. This is the most significant concern. Please refer to questions/concerns below for more details about this.

Overall, I enjoyed reading the paper. Moreover, the paper as a whole should be useful for the ML community. However because of some major doubts regarding discussion around COL which is the most crucial idea of the paper, I can only favour the paper for borderline acceptance currently.

**Questions:**

Questions:

(a) Loss weights: Among the loss weights to be set, $\gamma$ and $\lambda$ relating to COL seemed most crucial to me. The choice of $ \lambda=0.05 $ seems a little arbitrary. Could you explain how did you arrive at this value whether it be through a heuristic or otherwise? Also, is the system behaviour sensitive to choice of $\lambda$? How is the learning affected if its value changes? The experiment studying hyperparameters (appendix E) should also study role of $\lambda$.

(b) Statements surrounding COL: I got confused 2-3 times in understanding COL due to the statements made about it. Before going into the specific instances, I want to confirm that my understanding is accurate. Please correct me if I am mistaken in the following statements.
1. $q$ is a single feature representation used to predict **all** the concept labels.
2. In eq (5) $c_i, c_j$ denote concept labels for sample $i, j$ in a batch. These are binary vectors of size number of concepts. Consequently, COL encourages $q_i, q_j$ to be orthogonal if **ANY** concept label differs between samples $i, j$. COL also encourages $q_i, q_j$ to align if **ALL** concept labels among $i, j$ match.

If both these statements are correct, then these are my questions/concerns:
1. Is differing of any concept label between sample $i, j$ equivalent to them belonging to different classes (line 188, Pg 5)? Although this is a completely reasonable proxy but is it possible that two samples from same class differ in just one concept label and thus COL regards them as dissimilar?
2. Multiple times in line 182-186 in context of COL similarity loss, you state that it encourages similarity between feature representation for samples from "same concept". Isn't this an inaccurate phrasing? The accurate phrase should be for samples with "same set of concept-labels".
3. In line 290-292, you also refer to COL loss as "maximizing the inner product between the concept embeddings of different concepts". Firstly, it should be 'minimizing'? Also there is only one concept embedding $q$, right? Reading this statement combined with ones in previous point was particularly confusing as it conveys a meaning of orthogonality differently imposed copmared to COL. Namely, one could encourage different feature maps in $q$ to be used in prediction of different concepts, even for a single sample. This would more accurately represent "separating" representations of different concepts. But that is not what COL proposes.
4. Lastly, following the previous point, doesn't COL have some disadvantages in terms of imposing orthogonality for learning concepts? To be more specific, as an example consider two samples of similar classes. They are very likely to share many concepts (but not all). Isn't it sensible that the feature representation be similar as they share many concepts? The COL loss on the contrary encourages the feature representations to be completely dissimilar. Isn't it more sensible for dissimilarity loss to be applied **ONLY IF ALL** the concept labels present in two samples differ? If indeed this is the case, it needs to made very explicit.

**Limitations:**

The authors discuss the limitations in appendix. The content of the discussion covers some key points and is adequate for me.

---

> ### Author Rebuttal · Authors · 2023-08-09
>
> We would like to thank the reviewer for their feedback.
> > Loss weights: experiments around $\lambda$
>
> We experimented with different loss weights for $\lambda$ in our experiments and the model+COL seemed to be fairly robust with different values of $\lambda$. We have put those results in Table 5 of the rebuttal PDF on CUB and TIL datasets. While a fine-tuned value of $\lambda$ might show good performance, we observe that regardless, the model is still able to beat the performances of other baselines. We observed that 0.05 set as a good tradeoff between performance and uncertainty across datasets.
>
> > Definition of COL confusion.
>
> Upon second look, we can see why the definition of COL could lead to confusion, we would like to make a small change to the equation and update it. Hopefully, this can alleviate some of the concerns.
>
> $d_1=\sum_{\substack{i,j\in B,\\\\c_i^a = c_j^a\\\\a\in A}}\frac{q_i^T q_j}{||q_i|| ||q_j||};d_2=\sum_{\substack{i,j\in B,\\\\ c_i^a\neq c_j^a \\\\a\in A}} \frac{q_i^T q_j}{||q_i||||q_j||}$ where A is total concepts.
>
>
>
> > $c_i$ and $c_j$
>
> With the clearer definition, we would rename $c_i$ and $c_j$ indeed represent the binary concept vectors across the batch $i$ and $j$, while $c_i^a$ and $c_j^a$ are individual concept values for a given concept across the batch.
>
> >Is differing of any concept label between sample $i$, $j$ equivalent to them belonging to different classes (line 188, Pg 5)? Although this is a completely reasonable proxy but is it possible that two samples from same class differ in just one concept label and thus COL regards them as dissimilar?
>
> We understand that it might have been confusing if we look at Eqn 5, but looking at the revised version, we can see that it aligns with L188. Here the concept representation of two images from the same class is brought closer. It is also worth noting that datasets such as CUB and AwA2 have class-level concept representation.
>
> >Multiple times in line 182-186 in context of COL similarity loss, you state that it encourages similarity between feature representation for samples from "same concept". Isn't this an inaccurate phrasing? The accurate phrase should be for samples with "same set of concept-labels".
>
> Thank you for pointing this out. We will rephrase it as "concepts of the same class".
>
> > L290-292 typo. This would more accurately represent "separating" representations of different concepts. But that is not what COL proposes.
>
> We would like to apologize for the confusion. Indeed, L290-291 is a typo, it is minimizing not maximizing. Additionally, we would like to confirm that COL attempts to increase the separation between concepts by introducing the orthogonality constraint.
>
> > Isn't it more sensible for dissimilarity loss to be applied ONLY IF ALL the concept labels present in two samples differ? If indeed this is the case, it needs to made very explicit.
>
> Following your feedback, we have tried to make the COL definition more explicit mathematically. And dissimilarity loss, $d_2$ is applied in case of different image labels.
>
> We thank the reviewer for constructive feedback. We ran the experiments and attempted to answer all of the concerns of the reviewer and would be happy to answer any further questions.

---

> > ### Comment · Reviewer_VWed · 2023-08-15
> >
> > I want to thank the authors for the rebuttal.
> >
> > > With the clearer definition, we would rename $c_i$ and $c_j$ indeed represent the binary concept vectors across the batch $i$ and $j$, while $c_i^a$ and $c_j^a$ are individual concept values for a given concept across the batch.
> >
> > Your description about the new equation doesn't add more clarity. I think $i$ and $j$ are indices of samples and not the batch. Similarly $c_i$ and $c_j$ should be binary concept vector for specific samples and not the concept vector "across the batch".
> >
> > Your other references for $d_1$ and $d_2$, given below, give lot more clarity in this regard and align very close to my understanding.
> >
> > >  Here the concept representation of two images from the same class is brought closer.
> >
> > >  And dissimilarity loss, $d_2$ is applied in case of different image labels.
> >
> > This essentially leaves one of my questions unanswered which is about the application of dissimilarity loss $d_2$. One can have two samples with different class labels which share many (but not all) concepts, for eg. bird species with blue heads and small size in CUB. Your described training would still apply dissimilarity loss between the two samples since their class labels are different. Why is it sensible to apply dissimilarity loss in this case even though multiple concepts can be shared between the two classes? Would it not lead to very inefficient use of the feature representation $q$, since many feature maps will be detecting the same concept for different classes?

---

> > > ### Author Response · Authors · 2023-08-16
> > > **Response to the reviewer's question**
> > >
> > > We would like to thank the reviewer for their response. Please find the answer to your query below.
> > >
> > > Following your example, let us consider two different bird species, A and B with their respective binary concept vectors, $c_A = [1,0,1,0]$ and $c_B=[1,1,1,1]$. Now, let's suppose that these concepts denote attributes like "feathers," "beak," "wings," and "claws." We observe that the concept “feathers” and “wings” are the same for both samples. If our model were to learn shared feature representation, it would lead to softer (probabilistic) concept predictions. This, in turn, brings about what's known as concept leakage. This leakage fundamentally compromises the model's interpretability and trustworthiness because the concept predictor is no longer predicting the label on the basis of “hard” (0/1) concept information instead of probabilities. In the literature, [16,8] have observed that encoding of this soft information across concepts is not favored in CBMs since a supervisor cannot be sure if the model is predicting a concept because of its presence, or because it is encoding for something else. Additionally to limit the learning of correlations and leakage Havasi et al. [8] suggested that concept prediction must be independent. Hence shared representations are not ideal for concept representation despite their potential to improve task accuracy. Therefore in order to promote trustworthiness and robustness in concept explanation we propose to apply $d_2$ between $c_A$ and $c_B$. Using the dissimilarity cosine loss, we are able to create “disentangled” representations for concepts of each label.
> > >
> > > We hope that our answer cleared the reviewer's question and concern. We would be happy to answer any further concerns. Additionally, if we have answered your question, we would be grateful if you can kindly consider increasing the score.

---

> > > > ### Comment · Reviewer_VWed · 2023-08-17
> > > >
> > > > I would like some clarity on what information is intended to be independent and if $d_2$ ideally helps in achieving that in the current context.
> > > >
> > > > My understanding: When you refer to desiring concept prediction being independent to avoid leakage, I understand it as the information in feature representation $q$ used to predict labels for different concepts ("feathers", "beaks" etc.) to be independent, i.e. different feature maps in $q$ are used to predict the presence/absence of different concepts.  So ideally this would correspond to for example feature maps 1 and 2 in $q$ for predicting "feathers", 3 and 4 for predicting "beak", 5 and 6 for "wings", 7 and 8 for "claws".
> > > >
> > > > If bird species $A$ and $B$ did not share any concepts (for eg. $c_A = [1,1,0,0]$ and $c_B=[0,0,1,1]$), the loss $d_2$ would function perfectly fine encouraging feature representations used for $A$ and $B$ be orthogonal and thus different feature maps for different sets of concepts. However, if the species $A$ and $B$ share two concepts as in your example, encouraging orthogonality will still encourage use of completely different sets of feature maps in $q$ for samples from $A$ or $B$ which also carries some harms. For instance, the model might use feature maps 1-8 as above but only for samples from species $B$. The model will additionally use feature maps 9-10 for predicting "feathers" which activate for samples from species $A$ but not $B$ and similarly feature maps 11-12 for "wings" for species "A" but not "B". Use of feature maps 1-2 and also 9-10 for predicting "feathers" is the wastage I am pointing out. To avoid this wastage shouldn't cause any leakage or directly gain performance. My question is if a more considerate application of $d_2$ loss could help avoid any duplicating copies of feature maps to detect a concept while also preserving the disentanglement to great extent?
> > > >
> > > > Please feel free to correct any gaps or let me know if you have a different rationale.

---

> > > > > ### Author Response · Authors · 2023-08-18
> > > > > **Response to the reviewer's question**
> > > > >
> > > > > Thank you for your reply and explaining your concern. Please find the answer to your concerns regarding COL below:
> > > > >
> > > > >
> > > > > The dissimilarity loss $d_2$ encourages independent concept prediction. This is particularly important to reduce leakage in CBMs and improve the robustness of concept explanations. For the connection between the entanglement of concept predictions and leakage, we refer the reviewer to the introduction section on Page 2 of [A]. Motivated by this insight, we introduce $d_2$, to encourage disentanglement of the penultimate layer of features for concept prediction. We agree that this may result in an overcomplete representation, induced by the opposing forces of $d_1$ and $d_2$ for samples with partially overlapping concepts, and multiple feature groups may contribute to the same concept. However, it might be difficult to achieve disentanglement otherwise, and as our experiments show, COL improves the concept representation and accuracy, including in out-of-domain settings.
> > > > >
> > > > > To gain further insight into the effect of COL, we computed histograms of the activations of the penultimate layer (to which the loss is applied) and see that the histogram of CBM+COL is very sparse (with a large peak at 0 and a much smaller one at 1) in contrast to the vanilla CBM. This suggests that COL may encourage learning of overcomplete sparse feature space, the elements of which encode various combinations of concepts, and the last layer of $f$ learns to introduce invariance in the prediction of each concept $c_i$ with respect to specific combinations with other concepts $c_j$ by linear combination.
> > > > >
> > > > > Additionally, it must be noted that this regularization is only applied to the penultimate layer before concept prediction which means low-level features are still free to share weights. Essentially, we believe COL encourages learning a sparse over-complete dictionary of features with concepts still partially entangled in different combinations. The concept prediction layer in $f$ then learns a linear combination of these specialized features to introduce invariance with respect to the specific combinations.
> > > > >
> > > > > In fact, when studying histograms of activations of the penultimate layer, we observe that activations with COL are indeed very sparse in contrast to CBMs without COL. Given a sparse dictionary of combinations of concepts as induced by COL, the task of disentanglement of concept prediction would **ideally** reduce to a linear combination of dictionary elements. This is the intuition behind sparse coding (see for instance [B]). In our case, the sparsity is induced indirectly as a result of the orthogonality-based loss formulation.
> > > > > Our experiments show that this approach significantly facilitates the overall optimization of CBMs, improving concept accuracy and downstream performance.
> > > > >
> > > > >
> > > > > We hope that our answer cleared your questions regarding COL. We shall add your feedback and the above reasoning into the camera-ready version. We would be happy to answer any further concerns. Additionally, if we have answered your question, we would be grateful if you can kindly increase the score.
> > > > >
> > > > >
> > > > > [A]- Havasi, Marton, Sonali Parbhoo, and Finale Doshi-Velez. "Addressing leakage in concept bottleneck models." Advances in Neural Information Processing Systems 35 (2022)
> > > > >
> > > > > [B]- Lee, Honglak, et al. "Efficient sparse coding algorithms." Advances in neural information processing systems 19 (2006).

---

> > > > > > ### Comment · Reviewer_VWed · 2023-08-19
> > > > > >
> > > > > > Thank you for the response. It pacifies my concern to an extent.
> > > > > >
> > > > > > Taking into account your responses to all my concerns, I am raising my score from 5 to 6.

---

> > > > > > > ### Author Response · Authors · 2023-08-20
> > > > > > > **Thank you for your reply**
> > > > > > >
> > > > > > > We appreciate the fruitful discussion we had with the reviewer and their feedback. We will integrate your feedback and discussion about COL into the camera-ready version of the paper. We would be happy to address any further questions/suggestions that might come up until the end of the discussion period.

---

### Official Review · Reviewer_K2hb · 2023-07-04

**Soundness:** 2 fair
**Presentation:** 3 good
**Contribution:** 2 fair
**Rating:** 6
**Confidence:** 4

**Summary:**

This paper proposes Coop-CBM, a concept bottleneck model (CBM) trained using a novel multi-task loss and orthogonality regularizer. The proposed losses discourage undesired leakage in a CBM's concept representations while encouraging a better balance of their accuracy and interpretability. By incorporating these loss terms in their objective functions, *Coop-CBMs* are encouraged separate latent representations learnt for different concepts while reducing the intra-concept distance. This design enables effective concept interventions in these models while maintaining high concept and task accuracies. The authors perform an extensive evaluation of Coop-CBMs on four real-world datasets for image classification tasks, together with corrupted variants, and show that the proposed method achieves higher accuracy in all instances, compared to black-box models and previous CBM-based models, while remaining receptive to test-time concept interventions.

**Strengths:**

Thank you for the very well-written and interesting paper. I thoroughly enjoy reading this work, and I believe it is both addressing an interesting problem while having the potential to inspire different methods elsewhere. After carefully reading the paper and the accompanying appendix, I believe that its main strengths are:

1. Although the main idea in this paper is simple in nature, it is an interesting approach to alleviating some of the concerns that have been raised on the quality of learnt concept representations in CBMs, particularly regarding leakage.
2. The evaluation across four real-world datasets, as well as variations of these datasets with spurious correlations for OOD evaluation, makes the experiment section of this paper very well-designed. In particular, I believe there is a lot of potential impactful work in better understanding these models in OOD scenarios.
3. Given the need for interpretable architectures that are actually useful in practice, the core contributions of this work have the potential to be impactful if they stand thorough evaluation.
4. The main idea is not entirely novel, as it shares similarities to ideas previously proposed in disentanglement learning and contrastive learning. Still, its application to this space, and its evaluation in this space, is novel and worthy of study.

**Weaknesses:**

While I believe the ideas in this work have future potential, I think there is significant room for improvement in how this work is presented and evaluated. Specifically, I believe the following are its main limitations:

1.  I have some doubts about some of the claims made by the paper regarding the usability of its proposed method in concept-scarce (or concept-incomplete) settings. The results briefly discussed in passing in Section 5, and discussed in more detail in Section E of the Appendix, seem a bit suspicious to me without further clarification. This is because, intuitively,  one would not expect Coop-CBM to perform significantly better than CBM-AR or Joint-CBM when the number of training concepts is very limited (as the bottleneck's capacity will simply be too constrained!). Please see the question related to this point in the section below to see where my suspicion arises from. Please feel free to correct me if this is a misunderstanding on my end.
2. Another serious weakness in this work, which I believe other reviewers may also point out, is the number of sheer hyperparameters the proposed loss has ($\alpha, \beta, \gamma, \lambda$ for the loss and even more for the intervention algorithm). This. together with the lack of an accompanying ablation study showing whether the results presented in this work would change if one fine-tunes these hyperparameters as well as those in competing methods, severely limits the robustness of the evaluation. The claim that “for fair comparison, the concept prediction weight (here $\alpha$) in our baseline models is set to the same value”(lines 248-249) is in fact hard to take at face value. This is because it is unclear why all other hyperparameters are set to $0.01$ and whether this happens to be a value that greatly benefits the presented method. Therefore, more clarity on this end, including a potential ablation study and an appropriate hyperparameter selection for competing methods, would make the claims in this paper much stronger.
3. A lot of the details on the intervention side of things are left undefined in the main body of the paper and moved to the appendix. Currently, most of the contributions this work makes on the side of interventions (e.g., the study of user uncertainty or the use of a new intervention policy for selecting concepts) seem completely orthogonal to the rest of the paper’s work/main motivation and therefore are hard to evaluate within the context of the same paper. Don’t get me wrong: I believe these are interesting results; it is just that they currently seem out of place with almost all details pushed to the appendix (see how much methodological content section F has) and with a very limited evaluation if this is a core contribution. For example, it is unclear whether Algorithm 1 is used to select concepts in the results of Figure 1 of the main paper. Based on the caption saying that concepts are randomly intervened, I believe this is not the case. If it is, then it is not a fair comparison to compare Coop-CBM with a non-trivial concept selection policy against baselines that select concepts at random. If it is not, then there is no evaluation of the proposed concept selection policy anywhere in the main body of the paper. Either way, further evaluation of the proposed policy or clarification of how it fits in with the rest of the work is required.
4. I am a bit unsure about what the main takeaways of the experiments in Section 5.3 are that are not offered by previous experiments. Clarification on this end could help a lot.
5. The codebase provided is incomplete (I could not find some of the baselines) and missing key comments or pseudo-documentation. I am not expecting production-quality documentation in any way but at least a README to guide readers through it would be very useful.
6. Against traditional convention, and the authors' answer to the questionnaire question on error bars, Figure 1 and Tables 3, 4 and 5 are all missing error bars.
7. Although in the questionnaire the authors indicate the compute resources and licenses are stated in the paper/supplementary material, I was unable to find either of these.
8. Similarly, a lot of experimental/reproducibility details are missing from their codebase and their paper’s supplementary material, severely limiting the possibility of reproducing these results.

**Questions:**

After carefully reading the paper and the appendix, I had some trouble getting convinced by some of its results and proposed methods. My score is mostly influenced by this work's (1) lack of proper evaluation against existing baselines (specifically in how hyperparameters are selected), (2) its possibly counterintuitive results when the set of concept annotations is incomplete (and the lack thereof of an explanation for such unexpected results), and (3) the large amount of missing of details/explanations in some crucial components of this work that may lead to reproducibility issues. However,  I think my concerns may be addressed  by successfully addressing these questions:

1. If you look at Figure 3 of [1], you can see an evaluation of interventions in the method I believe this work calls “CBM-AR” (called “Hard AR w/o side-channel” in [1]). In that figure, it is clear that the task accuracy of CBM-AR is significantly low when one uses a small number of concepts during training (as one would expect as the model’s bottleneck is extremely constrained). However, in the results shown in Table 1 of the Appendix, the same drastic change in accuracy is not observed for CBM-AR (with only a drop of ~8% in task accuracy when using 10% of all concept groups). This makes me a bit suspicious of all of the results in this table. Therefore, I was wondering the following:
    1. When the paper says, for example, that 10% of concepts were used as annotations during training that means does that mean that the same group of 10% of training concepts were given for all samples of the training set **or** that, for any given sample, 10% of the concepts were selected at random and provided with annotations without regards of what concepts were provided to other training samples? That is, are concepts being subsampled at a "global" scale (where all samples are provided with the same subsample of concept annotations) or at a local scale (where each sample may have different concept annotations than other samples)? My guess is that it is the former (global scale subsampling) but I want to confirm this as it is unclear from the text.
    2. If it is the former as I believe, then is the paper using the side channel version of “Hard AR” when evaluating CBM-AR? If so, this should be made explicitly clear somewhere as no details could be found.
    3. **More importantly**, could you please clarify what is the intuition behind Coop-CBM having such a high accuracy when one uses a small number of training concepts (say 10%)? If the prediction used for the downstream task is that given by $g(f(x))$, then **I find this result extremely surprising**  given that Coop-CBM is still, just like its traditional CBM counterpart, severely constrained at the bottleneck when one uses a small number of concepts (and one would thus expect an even more severe drop in performance). One possible way I can see one can achieve this result is if the prediction used to compute the accuracy reported in Table 1 of the appendix is to use instead the immediate prediction made by $h$ (i.e., $h(f(x))$) as the output prediction of Coop-CBM. If that is the case, though, then this goes against how the model is originally defined in Section 3.2 and, more importantly, would mean that interventions are meaningless for this process as the prediction is bypassing the concept bottleneck. One convincing piece of evidence that your model is indeed working as expected (i.e., by predicting the label given by $g(f(x))$) is to show that it still positively reacts to interventions when the number of concepts is small (say 10%).
2. How do the results reported in some of the experiments change if one selects the hyperparameters for competing methods using a simple hyperparameter search (say over 0.1, 1, 10) for the concept loss weight in Joint-CBMs, CEMs, and CBM-ARs?
3. Could you please clarify the intervention concerns raised in the weaknesses? That is, how are interventions evaluated in Figure 1 and what is the relationship between these contributions and the rest of contributions? Are the contributions orthogonal to each other?
4. In equation (5), it is still a bit unclear what $q_i$ is. Could you please clarify this?
5. In section 5.1, why does 30% of the test set still have a spurious correlation? What is the rationale behind not removing the spurious correlation in its entirety to see how much the model truly depended on the shortcut for its predictions?

Regarding suggestions for the presentation, the paper is very well-written (thank you!). Nevertheless, the following possible errors and typos could be addressed before a camera-ready version:

- Typo in line 49: it should be “Mahinpei et al. [16] have shown” rather than “Mahinpei et al. [16] has shown”.
- Typo in line 96: parenthesis is accidentally underscripted in $f(c|x)$.
- Typo in line 127: missing “the” between “in” and “concept predictor”.
- Typo in equation (3): I believe the last term should be $\mathcal{L}_y(g(f(x)), y)$ rather than $\mathcal{L}_y(g(c), y)$.
- Typo in line 196: the sentence started in this line is a bit confusing (possibly missing a couple of words).
- This came out earlier this year, so I think it is ok to miss the reference. However, for the sake of completeness, it is worth noting that recent work [2] does consider human error/uncertainty when performing interventions as opposed to the claim in lines 299-300. This is not a big deal as it happened within a few months before NeurIPS’ deadline, but it is worth noting for a future camera-ready version of the manuscript.
- nit: ChatGPT and BARD are missing citations.
- nit: “XAI” in line 26 is never explicitly defined.
- The citation to “[1]” in the paper’s references of the main text is incomplete (the SENN paper reference).
- I couldn’t find anything related to the “Oracle Impurity Score” in the original CEM paper as cited in the Appendix in line 160. A quick search for that term found that it is defined in a separate paper [3].
- Section 3 could benefit from a diagram explaining the overall training process and architecture.
- This is pure nitpicking on my end, but I am still unsure what the “coop” part of “Coop-CBM” stands for as it is never explained/defined.

### References

[1] Havasi et al. "Addressing leakage in concept bottleneck models." *NeurIPS 2022*.

[2] Collins et al. "Human Uncertainty in Concept-Based AI Systems." *arXiv preprint arXiv:2303.12872* (2023).

[3] Espinosa Zarlenga et al. "Towards robust metrics for concept representation evaluation." AAAI 2023.

**Limitations:**

Although included in the appendix, this work provides a discussion of some of the limitations and potential impacts of its work. If possible, the manuscript could benefit from including such a discussion in the main body of the paper.

---

> ### Author Rebuttal · Authors · 2023-08-09
>
> We would like to thank the reviewer for very detailed feedback. We have updated the readme.
>
> > 1. 1. Clarification on sparse concept experiment.
>
> To provide clarity, when we mention "10%", it signifies that merely 10% of randomly chosen concepts were employed across all the images. Hence the same group of 10% of training concepts was given for all samples of the training set. This setup is designed to replicate scenarios involving sparse concept labels. This allows us to investigate the model's performance under conditions where concept annotations are limited, reflecting real-world scenarios of constrained concept availability.
>
> >1.2 Clarification on CBM-AR setup
>
> From Figure 3, Left image on CUB dataset, the red solid line Hard AR w/ side-channel is our baseline CBM-AR as the reviewer also alluded to. We shall make the setting of Hard AR more explicit in the camera-ready version.
>
> >1.3 Intuition behind Coop-CBM having such high accuracy in spare concept setting.
>
> * We would like to clarify that the task accuracy specified in the tables of the appendix and the main paper is g(f(x)) where the label prediction is bottlenecked on concepts only. We would like to build intuition using Figure 1 from the rebuttal pdf. Coop-CBM outperforms Joint-CBM due to its inherent design that introduces a "collaboration" between concept and supplemental task prediction at the same hierarchical level. In Coop-CBM, a task predictor stream is included alongside the concept predictor, enabling a synergistic relationship between the two. The concept features learned in Coop-CBM capture essential characteristics of the input data even when concept annotations are sparse. Through collaborative learning, Coop-CBM's concept predictor stream identifies and extracts meaningful relationships between the input features, labels and concepts.
> * In contrast, Joint-CBM despite being trained jointly follows a sequential approach where concepts are first predicted by the concept predictor stream and then utilized by the task predictor stream. This sequential approach can limit the capacity of the task predictor to capture subtle relationships in the data. Coop-CBM, by directly integrating the task predictor at the same hierarchical level, mitigates this limitation by combining both sets of information at an earlier stage, enabling better exploitation of the data.
> * We have plotted a graph for interventions for CUB in the rebuttal pdf.
>
> > How do the results reported in some of the experiments change if one selects the hyperparameters for competing methods [...]
>
> We compare the classification accuracy for CUB and TIL dataset across all of the baselines and hyperparameters mentioned. Due to lack of space, we weren't able to add generalization results. We observe that CEM and coop-CBM are fairly stable to diff hyperparams.
>
> >Could you please clarify the intervention concerns raised in the weaknesses? That is, how are interventions evaluated in Figure 1 and[...]
>
> We would like to clarify the reviewer's concerns by describing Figure 1 from the main paper.
> * L->R-The first two graphs show the importance of using human uncertainty(SCS) as a metric to select interventions as compared to concept uncertainty(CUS) and weightage(CWS). We show this in two ways:
> * First we plot for accuracy of the CBM model (as used by other baseline metrics [8,16]) when the annotator correctly intervenes on the concept. Intuitively, the accuracy should increase.
> * Second, consider that scenario is similar to the first except the annotator has changed. The annotator with ill intentions changes the concept values to incorrect concepts (instead of GT concepts). Now we plot all of the metrics and intuitively the accuracy would reduce.
> * L->R Last two graphs of Figure 1 compare the random interventions of different baseline models. Since SCS was present for CUB only, we consider random interventions for TIL and CUB. This also aligns with the evaluation method used in baseline (CBM-AR, CEM) papers.   We would also like to point out that this is a standard graph, also present in the baseline works. Hence we are not using any specific metric for coop-CBM, but we would like to emphasize that the concept representation of our model is enhanced due to COL. This enhancement is most evident in CUB.
>
>
> >In section 5.1, why does 30% of the test set still have a spurious correlation? What is the rationale behind not removing the spurious correlation in its entirety [...]
>
> 5.1 Experiment was inspired by the invariant learning/shortcut learning literature [1]. Here we expose the model to an extremely strong correlation, so a model purely minimizing training error will tend to exploit the color. But models will fail at test time because the background color setting is changed. For a naive model, if we change the spurious correlation to 0%, we will not be able to evaluate the performance well since a naive model will fail completely (would be random). Hence the literature changes the direction of correlation$\approx$1-strength during test time (usually between 20%-30%)
>
> >What the main takeaways of the experiments in Section 5.3 [...]
>
> 5.1 and 5.2 experiments modify the input image space while 5.3 changes the concept space. In 5.3, we introduce direct spurious correlation between the image and concepts (as compared to the image-label correlation earlier).
>
> >Compute and error bars.
>
> We have included compute in the rebuttal pdf. We shall include error bars from the missing tables in the camera-ready version.
>
> coop-CBM $\approx$ cooperative-CBM. We will clarify this in camera ready.
>
> We thank the reviewer again for the insightful comments. We hope we covered all concerns. We are happy to answer more questions during the discussion period.
>
> [1] Arjovsky, Martin, et al. "Invariant risk minimization." arXiv preprint arXiv:1907.02893 (2019).

---

> > ### Comment · Reviewer_K2hb · 2023-08-14
> >
> > Thank you so much for your thoughtful rebuttal. I have gone over it, and I appreciate the clarification of some of the doubts I had regarding missing details in the manuscript. Below I list a few leftover questions (marked as **Q**s) from your reply above:
> >
> > > when we mention "10%", it signifies that merely 10% of randomly chosen concepts were employed across all the images
> >
> > Thanks for clarifying this! To avoid future confusion, I would suggest being explicit about this if possible in the updated paper.
> >
> > >  the red solid line Hard AR w/ side-channel is our baseline
> >
> > Same as above, thanks for clarifying this! This is a non-trivial detail missing from the paper so I appreciate the promise to include it in the updated version.
> >
> > > The concept features learned in Coop-CBM capture essential characteristics of the input data even when concept annotations are sparse....
> >
> > I can somewhat see this, but I would appreciate it if further evidence was presented in favour of this hypothesis as it is hard to quantify a lot of the items in this argument. For example, the concept predictor in a Joint-CBM still receives feedback from the task labels even if this feedback comes implicitly via backpropagation, so I find it very surprising that the difference between joint CBMs and Coop-CBMs is as significant as the one shown in this paper. Not that this means that your presented intuition/argument is wrong in any way, but it is hard to be fully convinced by it without more evidence.
> >
> > For example, something I would consider to be evidence **against the argument offered** in the rebuttal is the following: In Figure 2 of the rebuttal pdf, it is shown that even when all concepts are being intervened on in both Coop-CBMs and Joint-CBMs, the accuracy of Coop-CBMs is much higher. If the true power of Coop-CBMs lies in it learning a better concept predictor due to its introduction of direct task feedback at the same hierarchical level, then this power should be almost entirely removed when one intervenes on **the entire bottleneck** as all activations generated by the concept encoder are overwritten by their ground-truth intervention values at this point. Therefore, it is very surprising and unexpected that even in this situation one sees Coop-CBMs being significantly more accurate than Joint-CBMs as their label predictors are operating on pretty much exactly the same bottleneck (i.e., there is no chance of extra information leaking from the concept encoder to the label predictor).
> >
> > **(Q)** Am I misunderstanding something? If so, I would very much appreciate it if the authors could clarify this. If not, then I would argue that there may be something in there that is not well-understood and that would benefit from further study.
> >
> > > We observe that CEM and coop-CBM are fairly stable to diff hyperparams.
> >
> > Thanks for including this! Could you please point me towards the table/figure and/or analysis for this ablation? I am sorry if I miss it in my previous reads. I found Table 7 in the appendix before, but this table is not mentioned anywhere in the appendix, or the main paper and I wasn't sure of the main conclusions from that table.
> >
> > **(Q)** Regarding my comment on this point in the weaknesses above, I would appreciate it if the authors could clarify why setting the concept weight across all methods to be the same is fair and whether this could potentially affect the results observed in this paper.
> >
> > > We would like to clarify the reviewer's concerns by describing Figure 1 from the main paper.
> >
> > Thank you for this. There may be a misunderstanding, but my concern was regarding some of the missing details on the evaluation rather than on the validity of the study performed. I agree that the way these experiments were run is standard and follow what has been done elsewhere in the literature. And I believe that if certain details are included in the main body of the manuscript, such as those raised in points 1.2 and 1.3 of the rebuttal, then this would strengthen the intervention-related sections.
> >
> > **(Q)** On Figure 1, on a third glance, I noticed that CEM's intervention accuracy in the last plot of Figure 1 for CUB is not as high as that reported in its original paper. Do you have an intuition as to why this is the case? I understand your CUB setup is slightly different as you have more concepts than that in the CEM and CBM papers. However, I am wondering whether the difference observed could be related to the lack of correct fine-tuning (e.g., $\alpha$) for baselines discussed elsewhere in my review. If so, then this may be a bit unfair for competing methods. Any further clarifications on this point would be much appreciated.
> >
> >
> > > Hence the literature changes the direction of correlation 1-strength during test time (usually between 20%-30%)
> >
> > This makes sense; thank you so much for clarifying.
> >
> > > We have included compute in the rebuttal pdf... error bars...
> >
> > Thank you, I appreciate this.

---

> > > ### Author Response · Authors · 2023-08-17
> > > **Thank you for your response**
> > >
> > > We would like to thank the reviewer for their thoughtful questions. We are happy to hear we were able to alleviate some of the concerns and questions by the reviewer. Please find answers to your further questions below:
> > >
> > > >Joint CBM vs Coop-CBM intervention
> > >
> > > Intuitively, this is likely due to Coop-CBM+COL minimizing the concept loss better (with help from the auxiliary loss which aids representation learning), which results in clearer separation of logits. Note, that the standard intervention protocol for CBMs replaces the concept predictions with the 5-percentile (for ground truth 0s) or 95-percentile (for ground truth 1s) of training logits. We verified via histograms that the logits of Coop-CBMs have a clearer separation with strong peaks at the beginning and end of the interval, without a wider cluster in between. Joint-CBMs on the other hand produce logits that have significantly more values in between, which in turn also moves the 5- and 95-percentile values closer together than for coop-CBMs. We will add this analysis and the corresponding logit-histograms to the paper. Please find below the 5- and 95-percentiles on the training set for both joint- and coop-CBMs:
> > > | Model        | avg 5th percentile | avg 95th percentile |
> > > |--------------|----------------|-----------------|
> > > | Joint CBM    | 0.25           | 0.83            |
> > > | Coop-CBM+COL | 0.03           | 0.92            |
> > >
> > > We believe this constitutes strong evidence in favor of this interpretation of the results.
> > >
> > > > Concept weights
> > >
> > > We have included the table of suggested concept weight parameter experiments (from the first review) in the global response, Table 6.
> > > As mentioned in the original paper, we set the concept weightage as 0.01. This hyperparameter is fixed across all of our experiments as the concept weightage. The initial intuition to select a number in this range came from Figure 2 of the CBM paper. Also, this is the value for which the results are reported for the CUB dataset in the original paper. Further, we performed some ablation studies regarding the same as noted by the reviewer in the appendix.
> > >
> > > > CEM results
> > >
> > > * We would like to tie this to the previous two answers as well. The CEM paper selects concept weightage for CUB = 5 (Pg 19, A.6)  which is the hyperparameter that is potentially the most divergent from our selected value. We tried our best to find the intuition behind the selection of $\alpha$ value or any related ablation study but could not find it in the CEM paper. This value is particularly interesting since they have used the same $\alpha$ value across their baselines, counterintuitive to the values used in the original CBM paper. Hence following their setup for comparison as well, we set the same concept weightage value (0.01) across the baseline.
> > > * As the reviewer also suggested that we have a slightly different setup in terms of a number of concepts. We would also like to note that our backbone is different, we use Inception V3 as the backbone across models.
> > > * Due to constrained time and resources, we conducted a small grid search for hyperparameters for CEM (0.01, 0.1, 1.0, 10.0) on CUB and we observed 0.01 gives the best task accuracy without much compromise in concept accuracy.
> > > * Finally, $\alpha$ hyperparameter particularly controls the behavior of a model under interventions which is more evident in CUB. In the adjacent graph for TIL however, we do not observe such a big disparity.
> > > * Similar to CBM, CEM, we use the same value across all of the experiments. Indeed changing a hyperparameter would change the results across any baseline but to the best of our knowledge, we have followed the regime of our baseline CBM and CEM papers. Additionally, we have performed ablation for the baseline methods to select the best operating point and all of them seem to converge to 0.01. We shall include these additional ablation results in the final version.
> > >
> > > We hope that the above points answered your question and concerns. We would be happy to answer any further concerns. Additionally, if we have answered your question, we would be grateful if you can kindly increase the score.

---

> > > > ### Comment · Reviewer_K2hb · 2023-08-18
> > > > **Thank you for your rebuttal**
> > > >
> > > > Thank you so much for the careful and well-written reply! Below I reply to some of the points made:
> > > >
> > > > > Intuitively, this is likely due to Coop-CBM+COL minimizing the concept loss better (with help from the auxiliary loss which aids representation learning), which results in clearer separation of logits.
> > > >
> > > > Alright, I can see this. My issue here is that I was under the assumption that a sigmoidal bottleneck was used for both models rather than a logit-based one. This key detail was missing from the paper and its supplement when I first read it. Thank you for this clarification. I would appreciate it if you could include these details, together with anything else necessary for the reproduction of results, as part of your updated manuscript.
> > > >
> > > > >  Note, that the standard intervention protocol for CBMs replaces the concept predictions with the 5-percentile (for ground truth 0s) or 95-percentile (for ground truth 1s) of training logits.
> > > >
> > > > I agree that this is how it is done for logit-based bottlenecks. However, as seen in the original CUB paper (see Figure 4 of [1]), it is known that at intervention time, using sigmoidal bottlenecks works much better as one intervenes more. A similar result in CUB was recreated in [2]. Therefore, I would suggest making this limitation explicit or including a sigmoidal joint CBM baseline as your comparison too.
> > > >
> > > > > We have included the table of suggested concept weight parameter experiments (from the first review) in the global response, Table 6.
> > > >
> > > > Sorry I missed this and thank you for including this! This adds some intuition on how this parameter affects your models.
> > > >
> > > > > The CEM paper selects concept weightage for CUB = 5 (Pg 19, A.6) which is the hyperparameter that is potentially the most divergent from our selected value. We tried our best to find the intuition behind the selection of value or any related ablation study but could not find it in the CEM paper.
> > > >
> > > > This is precisely why I pointed out the previous comment on CEM not matching the results observed in their paper (as the concept weight seems very different). I absolutely agree there seems to be an overview from the authors of the CEM paper when failing to include a rationale for the used parameters.  However, I would also argue that it is not entirely fair to compare against a baseline without using the configuration suggested by the authors of the baseline in CUB. I say this while fully understanding this was not done with ill intent and it does follow what others have done in the past. This is exactly what I am trying to avoid being perpetuated with my comments on this subject. Because of this, I would appreciate it if the camera-ready paper could explicitly state this possible unfairness in evaluation or include a baseline using the configuration recommended in the CEM paper for this model. I would guess that your model will still perform better (as CEM seems to struggle against a traditional joint-CBM with a sigmoidal bottleneck), so including such a baseline would strengthen your evaluation.
> > > >
> > > > ### Update to review
> > > >
> > > > The authors have addressed my questions and concerns, so I have increased my score to a (weak) accept. The reason for going for a weak accept vs something stronger is that (1) the large number of hyperparameters in Coop-CBM's training may limit the real-world application of this method and (2) the intervention policy contributions (as discussed in my review's weaknesses) are a bit orthogonal to the rest of the paper and lack depth compared to works that had proposed very similar things recently (as [2] which before ICML was public work from a NeurIPS workshop). Nevertheless, I do believe there is a lot of value in this work that is of interest to the wider community and I wish the authors the best of luck with this submission.
> > > >
> > > > ### References
> > > >
> > > > [1] Koh, Pang Wei, et al. "Concept bottleneck models." International conference on machine learning. PMLR, 2020.
> > > >
> > > > [2] Shin, Sungbin, et al. "A closer look at the intervention procedure of concept bottleneck models." arXiv preprint arXiv:2302.14260 (2023).

---

> > > > > ### Author Response · Authors · 2023-08-20
> > > > > **Thank you for your response**
> > > > >
> > > > > We would like to thank the reviewer for engaging in the discussion and for their feedback. We are happy to hear that the reviewer thinks that our work will contribute to the community.
> > > > >
> > > > > > Sigmoidal bottleneck
> > > > >
> > > > > We shall include a sigmoidal joint as a baseline in the paper as well. Thank you for the feedback.
> > > > >
> > > > > > CEM concept weightage choice
> > > > >
> > > > > As the reviewer also agreed that we followed the past papers. Regardless, we shall add results for CEM using their suggested concept loss hyperparameter.
> > > > >
> > > > > > the large number of hyperparameters in Coop-CBM's training may limit the real-world application of this method.
> > > > >
> > > > > We attempted to evaluate coop-CBM on various hyperparameter settings and observed its stability across different hyperparameters in the appendix (also performed during the rebuttal). We shall add further study to the paper. We hope that its robustness across distributional shifts with high concept accuracy motivates its real-world application.
> > > > >
> > > > > > the intervention policy contributions
> > > > >
> > > > > We proposed using a supervisor confidence score in addition to concept uncertainty and concept weightage (proposed by [2]). We also performed additional experiments (Appendix Figure 1, 2, 3) to evaluate the behavior of interventions under various distributional shifts (unexplored by the current literature) proposed by the paper.
> > > > >
> > > > > We would like to thank the reviewer for the interesting and detailed discussion which improved the quality of our work. We shall incorporate all of their feedback into our manuscript. We would be happy to address any further questions/suggestions that might come up until the end of the discussion period.

---

### Official Review · Reviewer_1MWB · 2023-07-06

**Soundness:** 3 good
**Presentation:** 4 excellent
**Contribution:** 3 good
**Rating:** 6
**Confidence:** 5

**Summary:**

The authors introduce an auxiliary task in CBM training to incorporate more information regarding the downstream task in the input representation for predicting concepts. They also introduce Concept Orthogonal Loss (COL) to enforce orthogonalization between input representations of different concepts, and vice versa. Experiments show superiority of their method over existing CBM baselines on the performance of the downstream task, for standard training settings as well as settings with manually-injected distribution shifts.

**Strengths:**

* Simple and straightforward presentation of the proposed methods. Very easy to read.
* Proposed method makes intuitive sense.
* Solid experiments.
* Experiments on extensive settings to demonstrate the superiority of CBMs over standard models w/o concepts.

**Weaknesses:**

* Risk of information leakage: Mahinpei et al. have shown that CBMs may leak information of the target task through the concepts, also mentioned in the paper (P2L50). Whether the performance gain of coop-CBM comes from information leakage is not exactly clear.
* Assumption of all concepts should be orthogonalized: The proposed COL implicitly assumes that all concepts should be orthogonal. The information gain shown in the experiments that this assumption holds true in the tested settings. In general, user selected concepts may not be entirely orthogonal. This method may be limiting when a user does not know whether adopting COL is appropriate given the chosen concepts.

**Questions:**

I am willing to increase the score and accept the paper as I genuinely believe this work is a good solid step in terms of making concept-based models more viable if the following issues are resolved properly.

### Risk of information leakage
My main concern is that coop-CBM improves downstream task performance because downstream task information is leaked through the concepts. This makes intuitive sense as the explanation of performance gain is the representation used to predict concepts contain more downstream task information. Actually I am not sure where the author stands on the topic of leaking information through the concepts. Does the authors believe that benign leakage of information is acceptable?

Perhaps experiments could be repeated by clipping the concept predicted values in coop-CBM to hard labels (as suggested in Mahinpei et al's paper) to reduce the leakage and see if there is still performance gain (or maybe this is already the default behavior indicated somewhere in the work that I missed).

The risk of information leakage is that then the concept scores no longer purely represents the concept, but rather encodes the downstream task label in some implicit manner. Then the whole concept bottleneck idea breaks down. The bottleneck is no longer a bottleneck and explainability through concepts is no longer valid.

### Assumption of all concepts should be orthogonalized
The authors extended the idea of concept whitening to CBMs through COL. However, orthogonalizing every concept may not be always beneficial (e.g. related concepts). Are the authors claiming orthogonalizing concepts is always beneficial? Is it possible to characterize when to adopt COL? Perhaps it is impossible to make definite statements about when COL would work but is it possible to at least provide a good heuristic that serves as a good-enough guideline?

For example, concepts in CUB are grouped into attributes of different parts of birds (e.g. body, head, wing). Does doing COL on inter-grouped concepts make sense? Perhaps something could be done with prior knowledge of which concept correlates more with others? Are there easy metrics that could serve as a proxies for concept correlation?

**Limitations:**

Yes.

---

> ### Author Rebuttal · Authors · 2023-08-09
>
> > My main concern is that coop-CBM improves downstream task performance because downstream task information is leaked through the concepts. This makes intuitive sense as the explanation of performance gain is the representation used to predict concepts contain more downstream task information[....]
>
> * We agree with the reviewer that information leakage is an important concern for concept-based models. As pointed out by the reviewer, and multiple times in our main paper, we have considered the impact of information leakage. This leakage occurs when the concepts that are supposed to be independent and interpretable become influenced by each other due to shared task-related information.
> * To overcome this issue, COL is designed to encourage orthogonality among the learned concept representations. Mathematically, COL adds an orthogonal regularization term to the loss function used during the training of concept classifiers. This term encourages the learned concept representations to be orthogonal to each other, thereby reducing any inadvertent inter-concept correlations.
> * One way to evaluate concept leakage in literature has been through concept accuracy.  COL enhances concept independence for concept-based models. In Table 2 (main paper), we evaluate the concept accuracy of each baseline model on the CUB dataset. Here, as one can expect the “hard” CBM has the highest concept accuracy in the absence of COL. But with the introduction of COL, the concept representation for each baseline was learned to reduce the information leakage. Coop-CBM+COL achieves comparable concept accuracy to Independent (hard) CBM with higher mean and comparable uncertainty.
> * This result aligns with the fundamental goals of explainability and interpretability, which are pivotal in ensuring that the concept-based model's representations remain meaningful and free from unintended correlations, ultimately enhancing the model's utility and reliability in various applications.
>
> >Perhaps experiments could be repeated by clipping the concept predicted values in coop-CBM to hard labels (as suggested in Mahinpei et al's paper) to reduce the leakage and see if there is still performance gain (or maybe this is already the default behavior indicated somewhere in the work that I missed).
>
> Thank you for suggesting the experiment. Following your suggestion, we performed two sets of experiments on CUB and TIL datasets using coop-CBM.
> * First, we trained the model by clipping the predicted concept values to “hard” labels (Table 3, Exp1 in rebuttal pdf).
> * Second, we trained the model as we have described earlier in the paper (using soft labels) and the model evaluates on the test set by clipping to “hard” labels (Table 3, Exp2 in rebuttal pdf).
> From the above experiments, we conclude that the model is able to learn a good representation of the concepts without necessarily leaking information.
>
> >Are the authors claiming orthogonalizing concepts is always beneficial? Is it possible to characterize when to adopt COL? Perhaps it is impossible to make definite statements about when COL would work but is it possible to at least provide a good heuristic that serves as a good-enough guideline?
>
> We would like to connect this answer with the previous answer. The motivation for COL arose to mitigate information leakage in coop-CBM and is applicable to other concept-based models too. The rationale for COL revolves around its ability to counteract the inadvertent propagation of task-specific information across concept representations. By imposing this constraint, COL effectively disentangles the influence of different concepts, fostering their independence and reducing the risk of cross-concept information leakage interpretability, and the effect of interventions. Also, please see below for detailed answers.
>
> >For example, concepts in CUB are grouped into attributes of different parts of birds (e.g. body, head, wing). Does doing COL on inter-grouped concepts make sense? Perhaps something could be done with prior knowledge of which concept correlates more with others? Are there easy metrics that could serve as a proxies for concept correlation?
>
> * Devising an optimal point for when to use a loss function could be great future work building on our work. Also, prior knowledge about concept groups cannot be assumed. One way to characterize the correlation could be through using mutual information, which can be an interesting followup work.
>
> * We would like to emphasize that COL can be used to improve the concept representation of any concept-based model. For instance, considering the case of Joint-CBMs, which has been known to exhibit information leakage, the incorporation of COL can effectively increase separation in concept learning leading to enhanced interpretability while mitigating the risk of information leakage (See Table 2 main paper).
>
> * In order to investigate the effectiveness of COL we propose a comprehensive experiment that deliberately explores an extreme case for evaluation in line with the reviewer's proposal. We aim to showcase the robustness of COL across different scenarios, extending the OOD experiments in the main paper.
> Experiment- We consider a scenario where input concepts are intentionally duplicated to create a high degree of concept correlation. In our experiment from Table 4, we duplicated 10%, 25%, 50% and 100% of concepts and added them to the original concept bank. This is a worst-case representation of “similar concepts”. From the table, we see that the duplication of concepts does not impact the concept or the task accuracy.
> Additionally, this experiment contributes to the broader understanding of how COL performs in various scenarios.
>
> We thank the reviewer again for the detailed questions and insightful additional experiment suggestions. We hope we addressed all concerns and are happy to answer and further questions during discussion period.

---

> > ### Comment · Reviewer_1MWB · 2023-08-18
> >
> > Many thanks to the authors for the additional experiments. Those were quite insightful and definitely proved a couple of points.
> > Before examining the experiment results, based on the authors' response, it seems like the authors' consider concept correlation as a cause for information leakage and thus COL as a cure.
> > > This leakage occurs when the concepts that are supposed to be independent and interpretable become influenced by each other due to shared task-related information.
> >
> > I don't believe that to be true. Consider the following toy scenarios for illustration:
> > 1. The target prediction $y$ is one of the concepts $c_i$. Even if we orthogonalize the concepts mutually, the information of $y$ is still leaked through $c_i$. &rarr; Concept mutual orthogonalization does not imply no leakage of target information.
> > 2. The concepts and target are completely independent. Let us duplicate concept $c_i$ to be $c_i'$ and add that to the concept set. Then the concepts are not mutually orthogonalize but no information leakage is happening. &rarr; No leakage of target information does not imply concept mutual orthogonalization.
> >
> > Following the arguments, concept mutual orthogonalization and target information leakage are two completely different conditions and there are no direct mutual implications. Please correct me if the authors' believe this to be untrue.
> >
> > In that case, COL is not a cure for information leakage and concept disentanglement is not a proxy for measuring information leakage. Instead, information leakage should be measured with mutual information between the concepts and target label directly.
> >
> > Now moving on to the experiments, Table 3 showed the experiments with predicted concepts clipped to hard labels and we do observe a performance drop compared to the non-clipped version. Particularly for CUB, the performance of Coop-CBM dropped to lower than CEM. This suggests that if we apply some preventative measures for information leakage, Coop-CBM would perform similarly to the non-coop version. The direct implication is the performance of Coop-CBM may come from information leakage. It would be a stronger argument if the authors also present the results for AwA2.
> >
> > For Table 4, the authors' conducted a very interesting experiment on how duplicating a concept (the worse case scenario for correlated concept) affects the performance of Coop-CBM. I must applaud the authors' for putting their method under such scrutiny, experimenting in such extreme scenarios. As the authors have stated, the performance of Coop-CBM maintains at a similar level despite how much duplication is happening. At first glance, this seems to imply the robustness of the method. But if we think about the implications, these duplicated concepts are completely correlated and then de-correlated with COL. How is it possible for completely dependent concepts be de-correlated? Well, correlation is linear so as long as the latent dimension is large enough, we could find enough orthogonal directions to represent the same concept. This suggests that even if we apply COL and the disentangling loss suggest the concepts are disentangled, the concepts may still be dependent. In this case, completely dependent (duplication). This is quite alarming. Correlation is not independence. De-correlating does not necessarily remove the dependence between concepts (in the probabilistic sense).

---

> > > ### Author Response · Authors · 2023-08-19
> > > **Response to the reviewer's detailed feedback**
> > >
> > > We thank the reviewer for their detailed feedback. Please find our comments below.
> > >
> > > Initially, we’d like to point out that we don’t claim that COL leads to 100% leak-proof model but encourages disentanglement between concept predictions. This results in more accurate concept predictions, suggesting more trustworthy explanations. Seminal work on leakage [8] (CBM-AR baseline) uses concept accuracy to evaluate leakage in CBMs which does not achieve full accuracy on concepts either. Also, we believe that limiting the discussion to the two extreme cases brought up by the reviewer obscures important nuances.
> > > >Regarding 1)
> > >
> > > For $y=c_i$ and information only passing through $c_i$, we disagree with the notion that this should be termed leakage. In this case, $y$ could be explained in terms of concept $c_i$, as intended.
> > > >Regarding 2) and the topic of co-occuring concepts
> > >
> > > COL does not yield perfectly orthogonal representations, as the reviewer suggests, but encourages a degree of disentanglement between concepts. In the discussion with other reviewers, we also presented an interpretation with supporting evidence that this might be due to learning of an overcomplete sparse representation. In particular, COL does not achieve perfect disentanglement between co-occurring concepts. However, given an overcomplete representation capturing different combinations of a concept $c_i$ with others, the concept prediction layer can introduce invariance of the concept prediction wrt specific combinations by learning weighted sums over multiple differently entangled features. This improves concept accuracy, enabling task predictions based on concepts. For details please refer to reviewers-VWed, K2hb answers. Specifically, we analyzed learned representation and concept predictions using histograms. In the analysis for reviewer VWed, we show that COL induces a very sparse representation. In the analysis for reviewer K2hb, we show for coop-CBM+COL that resulting concept predictions are more separated with clear peaks at 0 and 1.
> > > >Regarding Table 4
> > >
> > > The intermediate classification loss (coop-) aids representation learning, while COL encourages disentanglement (only between non-identical concepts) in the representation. As described above, this enables the introduction of invariance of individual concept predictions wrt other concepts. This improves concept accuracy and the downstream model can rely on these more accurate concepts, instead of leakage as seen in Table 4, which also explains why coop+COL enables more effective interventions (see the discussion with reviewer K2hb).
> > > >Regarding mutual information
> > >
> > > We would like to point out that we already mentioned that MI could be investigated in Appendix H. Experiments from [35] may suggest that there could be potential a correlation between concept accuracy and the MI plane (needs exploration!). We use concept accuracy, a widely accepted metric for leakage, as used in works that address leakage [8,16]. [8] uses MI to evaluate the completeness of side-channel different from leakage. The connection between independence of concept predictions and leakage is discussed on Page 2 of [8] which motivated our work on COL.
> > > >Regarding Table 3
> > >
> > > Experiments on AwA2 had not finished earlier. Please find the results below.
> > > |          | Std         | Exp1        | Exp2        |
> > > |----------|-------------|-------------|-------------|
> > > | Coop-CBM | 96.6+-0.1 | 96.3+-0.2 | 96.5+-0.1 |
> > > | +COL     | 97.0+-0.1 | 96.7+-0.2 | 97.0+-0.2 |
> > >
> > > In Table 3, we conduct two experiments, where the performance of Coop-CBM (without COL) is slightly lower than that of CEM (NB comparing standard training of CEM not clipped) only on the CUB dataset for Exp1. Despite that, and as mentioned in our paper, COL improves concept and label representation. Hence, using clipping as a preventive measure for leakage along with COL, our model significantly outperforms the baselines including CEM. Further, we now highlight the difference in performances between Exp1 and Exp2. In Exp1, we round soft concept probabilities to 0/1 during training, while in Exp2, we round concept values to 0/1 during test-time. The bigger drop in performance in Exp1 could also be explained by non-smooth changes in the input distribution of $g$ during training when concept predictions cross the boundary of 0.5. Regardless, we would like to point out that coop-CBM performs on par with standard black-box models, while also providing explanations for CUB and surpassing the performance of baselines incl. black-box models on TIL and AwA2. And with the addition of COL, we surpass all baselines on all considered datasets, despite constraining the model to hard concepts both during training and evaluation.
> > >
> > > We hope that our answers and comparison (suggested by the reviewer) would be able to alleviate the reviewer’s concerns. Additionally, if we have answered all of your questions, we would appreciate it if the reviewer would consider accepting the paper as mentioned.

---

> > > > ### Comment · Reviewer_1MWB · 2023-08-19
> > > >
> > > > Thank you authors for the detailed responses and additional experiments on AwA2. It seems like this whole argument is built on the claim made in [8] that concept accuracy is a good metric for measuring leakage.
> > > > > We use concept accuracy, a widely accepted metric for leakage, as used in works that address leakage [8,16].
> > > >
> > > > I tried looking for the claim in [8] but couldn't find it. Could the authors kindly point out the corresponding section?
> > > >
> > > > --------------------------------------------------------------------
> > > >
> > > > Given the good experiment results when clipping to hard labels imply that maybe this method does work. To avoid risking information leakage, shouldn't the method directly adopt the clipping? As it only lowers the performance by a little, which likely comes from concept leakage, it seems to only make sense to make the tradeoff.

---

> > > > > ### Author Response · Authors · 2023-08-20
> > > > > **Thank you for your feedback**
> > > > >
> > > > > We would like to thank the reviewer for the detailed follow-up. We try to clarify the details below.
> > > > >
> > > > > There are several statements in [8] which taken together suggest that concept accuracy is a good proxy for leakage mitigation, e.g. the last paragraph of Section 4.1:
> > > > >
> > > > > > The side-channel model [their proposed method] should still be used to avoid the detrimental effects of leakage on concept accuracy[...]
> > > > >
> > > > > or the second paragraph in Section 5.2:
> > > > >
> > > > > > Soft CBMs underperform in concept accuracy, but they are often able to predict the label correctly (Table 1). This is only possible if the concepts are leaking information about the label in an uninterpretable way.
> > > > >
> > > > > Our take on the connection between concept accuracy and leakage mitigation is the following:
> > > > >
> > > > > 1. A high concept accuracy allows the downstream classifier $g$ to rely more on accurate concepts for label prediction.
> > > > >
> > > > > 2. Replacing the predictions of a highly accurate concept predictor via interventions does not introduce values far outside the input distribution that $g$ adapted to during training, allowing a higher gain in label accuracy.
> > > > > 3. If concept predictions are closer to the ground truth binary values, the bottleneck is also tighter than for less restricted continuous vectors, limiting the capacity for encoding of additional information (i.e. leakage).
> > > > > 4. Admittedly, for a complete picture, one may have to look at both label accuracy and concept accuracy, since lower concept and lower task accuracy together (in comparison with hard-CBMs) may also be indicative of low leakage or $g$ not exploiting a leak for higher task accuracy. However, because for coop-CBMs+COL task accuracy is consistently higher than the baselines, we focus on concept accuracy in our discussion on leakage mitigation. Essentially, concept accuracy alone may not be a good metric for leakage mitigation, but given high task accuracy it becomes an appropriate proxy for it. Our approach improves both concept and label accuracy compared to the baselines on CUB, TIL and AwA2 datasets.
> > > > >
> > > > > > Given the good experiment results when clipping to hard labels imply that maybe this method does work. To avoid risking information leakage, shouldn't the method directly adopt the clipping? As it only lowers the performance by a little, which likely comes from concept leakage, it seems to only make sense to make the tradeoff.
> > > > >
> > > > > We would like to thank the reviewer for suggesting insightful experiments by clipping the concept values. We shall add the results presented during the rebuttal of clipping concept values both during training and test-time into the manuscript. We agree that it makes sense to integrate clipping into the overall method and we will add this suggestion in the manuscript.
> > > > >
> > > > > We hope the above discussion clarified our motivation and reasoning and has addressed your concerns. If so, we would appreciate it if you could kindly increase the score.

---

> > > > > > ### Comment · Reviewer_1MWB · 2023-08-21
> > > > > >
> > > > > > Regrading the part about a tighter bottleneck being less likely for encoding information, merely encoding however many classes for the target downstream task is enough to leak uninterpretable information. Thus, I don't think concept accuracy would be an accurate proxy for information leakage. But since the proposed method works when the predictions are clipped to hard labels, my current understanding is that the coop-part of training provides a relevant auxiliary task that helps the learning of concepts (similar to the inductive bias for multi-task learning), which according to my understanding is the main reason Coop-CBM works.
> > > > > >
> > > > > > For the disentangling part (COL), the discussion with Reviewer VWed covered most of my concerns.
> > > > > >
> > > > > > As the authors have agreed to integrate concept clipping into the main method to prevent leakage and I have no other major issues, the score is increased from 4 to 6. I would opt for a stronger accept if there were more design ablation (e.g. studying what types of concepts would benefit from COL, since it is quite intuitive that some concepts wouldn't). Nevertheless, this is a solid technical paper for training CBMs better.

---

### Official Review · Reviewer_Mi12 · 2023-07-10

**Soundness:** 3 good
**Presentation:** 3 good
**Contribution:** 2 fair
**Rating:** 6
**Confidence:** 3

**Summary:**

The author proposes coop-CBMs + concept orthogonal loss (COL) to solve previous limitations in Concept Based Models with a focus to learn relevant concept representations that consecutively boost model performance.

**Strengths:**


In general, the idea of coop-CBMs+COL is simple and effective. The method increases prediction accuracy on benchmark datasets and  seems to work well on noisy inputs.

**Weaknesses:**

* The overall structure of the paper is reasonable, but part of substructure in some sections are not well organized. In introduction section, the author first introduce definition of previous Concept bottleneck models, then pointing out its limitation.Then followed with the importance of studying the robustness of CBMs in various domains. After that, returns back to bottleneck of CBMs. Unless the author aims to describing totally different limitations of previous CBMs, it feels unnecessary.

* In Section3, the author does not specify what does $\hat{\theta}$,  $\hat{\Phi}$ stands for,  it would be better to insert formula into the words such as a formula followed by it notations. Otherwise this description looks messy.

* The author does not show experimental results for using CBM alone on classification Coop-CBM.

* It's confusing calling Coop-CBM both a component of the final strategy and the whole strategy incorporating it.There is a bit confusing that what is your final method. Coop-CBM or Coop-CBM + COL ?




**Questions:**

1. modify the structure/content arrangements in introduction section.
2. do more literature review on section2 on the relationship between Concept-Based Model and explainability if applicable.
3. please clarify the notations in method part.
4. please clarify what is your final method, Coop-CBM or Coop-CBM + COL ?
5. add classification results for only using Coop-CBM on CUB, AwA2 and TIL datasets.
6. better to tell the significance of the proposed method, not just limited to some improvements in the benchmark dataset.

**Limitations:**

No.

it would better to discuss the limitations of proposed methods ( Coop-CBM along with COL) in `Discussion and Conclusion` section.

---

> ### Author Rebuttal · Authors · 2023-08-09
>
> We would like to thank the reviewer for their feedback.
> > The overall structure of the paper is reasonable, but part of substructure in some sections are not well organized.
>
> Thank you for your feedback. We shall rewrite that section by moving the two paragraphs about distributional shifts and limitations of CBMs around.  We will update the camera ready version accordingly to improve readability.
>
> > the author does not specify what does $\hat{\theta}$, $\hat{\phi}$ mean.
>
> $\hat{\theta}$, $\hat{\phi}$  are the parameters of the encoder and $g$ correspondingly (please see Figure 1 from rebuttal PDF).
>
> > The author does not show experimental results for using CBM alone on classification Coop-CBM.
>
> In our initial submission, we have already conducted comprehensive evaluations of CBM, coop-CBM, and the inclusion of COL, (please refer to Tables 1, 3, 4, and 5 in the main paper, along with Tables 1, 2, 3, 4, and 5 in the appendix). Furthermore, we have conducted an additional analysis in Table 2 by isolating the effect of COL within different baseline models. This analysis demonstrates that regardless of the chosen concept model, the incorporation of COL consistently enhances concept accuracy.
> > It's confusing calling Coop-CBM both a component of the final strategy and the whole strategy incorporating it.There is a bit confusing that what is your final method. Coop-CBM or Coop-CBM + COL ?
>
> As per the title of the paper, we are suggesting losses that can be used to improve the generalization of concept-based models. We will add a sentence in the introduction to emphasize this. Our contribution in terms of the model is two fold: first to improve task accuracy for concept-based models via Coop-CBM, and second to improve the concept accuracy for concept-based models via COL. We empirically show that each of the strategies alone works but definitely the most effective strategy is to use Coop-CBM + COL. From Table 1,3,4,5 penultimate line, we observe that Coop CBM alone improves task accuracy but with COL, we can further enhance the task accuracy. As mentioned in our paper, information leakage is detrimental to concept accuracy and hence the explainability of the model. We thus add the COL loss to increase the independence of learning each concept from the other as observed in Table 2 of the main paper for CUB.
>
> > do more literature review on section2 on the relationship between Concept-Based Model and explainability if applicable.
>
> We did our best to include all of the recent related references of Concept-Based models that were released until the Neurips deadline in the related works section of the main paper. We will add all new references mentioned by other reviewers. Please let us know if you think we are missing a particular reference and we will include it in camera ready. In the Appendix, section D, of the original submission we have included a further literature review on explainability.
>
> > please clarify what is your final method, Coop-CBM or Coop-CBM + COL ?
> Our contribution encompasses the introduction of both coop-CBM and COL. Both these methods, individually and when combined, exhibit improvements in both explainability, accuracy and test-time interventions.
>
> > add classification results for only using Coop-CBM on CUB, AwA2 and TIL datasets.
>
> The main paper has classification results for using coop-CBM on CUB, AwA2 and TIL datasets in Table 1. In addition, we have results for coop-CBM only on all of our other tables as well in the original paper. The penultimate row of each of the tables mentioned that says “Coop-CBM” signifies classification results using only Coop-CBM. The “+COL” row signifies the classification result using both Coop-CBM+COL. We will add a sentence to each table caption to highlight this.
>
> > better to tell the significance of the proposed method, not just limited to some improvements in the benchmark dataset.
>
> This work introduces two significant contributions to concept-based models. Firstly, we propose a multi-task model that predicts an intermediary task label alongside concept predictions. This is particularly useful when dense and relevant concept annotation is lacking, as seen in the TIL dataset. Secondly, we incorporate orthogonality constraints in the concept representation space through concept orthogonal loss during training. This loss enhances inter-concept separation while reducing intra-concept distance. Through extensive experiments across diverse datasets and distributional shifts, we observe that the bottleneck layer preceding the final prediction enhances the robustness of concept-based models against spurious background correlations. Coop-CBM combined with COL demonstrates leading performance in both task and concept accuracy.
>
> Additionally, we would like to point out that most of the current CBM works do not perform extensive experiments on distributional shifts. Our paper is the first work that attempts to shed light on the impact of using CBM on multiple different distributional shifts. We have attempted to justify and motivate our work both empirically and intuitively. In Section 3, we introduced the intuition behind our contributions and then throughout the paper we provide analytical experiments which highlight each contribution’s effectiveness.
> Please point us to any specific issues and we will address them during the discussion period.
>
> > it would better to discuss the limitations of proposed methods ( Coop-CBM along with COL) in Discussion and Conclusion section.
>
> We have included limitations in Appendix B. Following your suggestion we will move this discussion to the Conclusion for camera-ready version.
>
> We have attempted to answer all of the concerns of the reviewer and would be happy to answer any further questions. We thank the reviewer again for their feedback.

---

> > ### Comment · Reviewer_Mi12 · 2023-08-17
> > **Acknowledgement of rebuttal**
> >
> > Thank you for the detailed response and the additional clarification. The rearrangements in details did make this paper with stronger evidence. Also, the clarification make writing more clearer and easier to understand your proposed Coop-CBM.
> >
> > Another minor concern is that Do you use the same training strategies for datasets in table 2 and table 3 (I mentioned in-domain dataset here)? If so, what's the difference between accuracy result from in-distribution results in table3 and the one from table2 ?

---

> > > ### Author Response · Authors · 2023-08-17
> > > **Thank you for your response**
> > >
> > > We are grateful for the feedback provided by the reviewer that strengthened our paper, and we are glad that our reply clarified most of the reviewer’s doubts. With regards to training arrangements in Table 2 and Table 3, please find our answer below:
> > >
> > > * Using Table 2, we observed that our proposed loss COL can be applied to any baseline method to improve their concept accuracy. We would like to point out that Table 2 is not part of the OOD experiment as mentioned in the corresponding text.
> > >
> > > * In Table 1, we evaluated the task accuracy (classification label prediction) of the baselines and our proposed model on the test splits provided by the datasets. Similarly, for Table 3 we are evaluating the task accuracy. In Table 3, as the reviewer alluded to and mentioned in the text, we create synthetic datasets to evaluate the performance of models in the presence of spurious correlation.
> > >
> > > * In the synthetic datasets, we spuriously correlated the background with the label for the CUB dataset and hair colour with gender for the Celeb A dataset. Let us take the CUB dataset for example, where we segment the foreground of each image and add a coloured background. We generate 200 background colours and correlate each with one of the 200 class labels. Now an image from the in-domain test set follows the same distribution as the training image i.e. spurious correlation present, but in the out-domain test set, the strength of this correlation is reduced, strongly impacting the accuracy. The strength of the correlation is 80% for in-domain train and test and 30% for out-domain test. We have included an example for in-domain train, in-domain test and out-domain test in our supplementary material in Figure 4.
> > >
> > > * This synthetic dataset, constructed specifically for contrasting in- and out-domain performance, is different from the original CUB dataset, which is why the in-domain test performance also differs.
> > >
> > > * The hyperparameters for each model are kept fixed across experiments in Table 1, Table 2 and Table 3.
> > >
> > > We hope that our answer appropriately addressed the question and concerns. We would be happy to answer any further concerns. Additionally, if we have answered your questions, we would be grateful if you would consider increasing the score.

---

> > > > ### Comment · Reviewer_Mi12 · 2023-08-18
> > > > **Acknowledgement of rebuttal**
> > > >
> > > > Thanks for your response. Based on your explanation, I have no other concerns, please make sure all the rebuttal results are added to the camera ready version, and I would like to increase the rating to acceptance with rating 6.

---

> > > > > ### Author Response · Authors · 2023-08-18
> > > > > **Thank you for your response**
> > > > >
> > > > > We appreciate the feedback given by the reviewer and are pleased that our response effectively answered your concerns. Your feedback has enhanced the quality of our work. We will integrate your feedback into the camera-ready version of the paper. We would be happy to address any further questions/suggestions that might come up until the end of the discussion period.

---

### Official Review · Reviewer_G8F4 · 2023-07-10

**Soundness:** 2 fair
**Presentation:** 2 fair
**Contribution:** 2 fair
**Rating:** 6
**Confidence:** 4

**Summary:**

This paper proposes auxiliary losses to learn concept-bottleneck models: a coop task loss, and a COL (orthogonal concept loss). The experiments show that the proposed coop-cbm can perform 1-3% better than standard black-box models and outperform other CBM models on three datasets: CUB, AwA2 and TIL.

==================
post-rebuttal: the authors have addressed my concerns and questions in the rebuttal. I would like to increase the rating from 4 to 6.

**Strengths:**

1. This topic is important
2. The paper compared with 3 fairly new CBM baselines (CBM [12], CEM [35], CBM-AR [8]), but stronger baselines are not compared [33, 18]. The authors mentioned it's an unfair comparison to [33] in Line 228-229, but it's still not clear why, and why [18] is not compared either.

**Weaknesses:**

1. The proposed idea is interesting, though the two losses (COOP loss Eq 3, and COL in Eq 6) seem to be proposed in prior works? The authors mentioned that COL is from [21], and it feels like the COOP loss in Eq 3 is similar to the non-interpretable "residual" terms in Post-hoc CBM [33].  So the main contribution seems to be integrating them together? Please clarify and compare if there's some misunderstanding here.

2. The description of some related work seems not accurate: Line 74-76: what does it mean that "suffer from pretrained model's biases"? Post-hoc CBM [33] with residual terms can improve the downstream task accuracy. It looks like COOP can perform better is due to the residual loss + join training (since post-hoc CBM only training the last 2 layers)? Besides, the result in Label-free CBM [18] show that it is comparable to original downstream task performance without the residual terms. Please compare with [33] and [18] with ablation study.

3. Large scale datasets results are missing (ImageNet)

4. The exposition of the technical part can be improved (sec 3). For example, to describe f, g, h, it'll be useful to visualize the structure. phi and theta are not defined. Also, since f and h actually differ in one linear layer, why both of them still use theta? it seems phi is the parameter of g.




**minor**
- an extra s in Line 277 "modelss"

**Questions:**

1. It's not clear why the concepts are necessary need to be orthogonal. Could the authors explain?

2. In eq 5, how do you define intra-concepts? It looks like you are using c_i= c_j, which means the concepts are exactly the same? Please elaborate the details.

3. Line 228-229, why is it unfair to compare with post-hoc CBM [33]? Do you mean that it is expected that [33] with joint training can outperform coop-cbm? please clarify

4. Please report computation cost for baselines and the proposed method in Table 1.

5. How is Fig 1 calculated? Is it correcting the mis-classified images in the test time through intervention? Please give more details on how Fig 1 is plotted.

**Limitations:**

I did not see a dedicated paragraph to discuss the limitation of the proposed work.

---

> ### Author Rebuttal · Authors · 2023-08-08
>
> We would like to thank the reviewer for their feedback.
> >The authors mentioned it's an unfair comparison to [33] and why [18] is not compared either.
>
> [18] and [33] used a pre-trained model - CLIP which was trained on a massive corpus of data to obtain concepts. This can potentially introduce inherent biases from pretraining into the concepts. This was also brought up in the Limitations and Conclusion section of [33]. Furthermore, the dissimilarity in the concepts employed in these works adds complexity to establish a fair and meaningful comparison. Moreover, we wish to **emphasize** that neither of these works directly compare with CBM variants in their main paper, except for [33] which appears in Appendix C. Also it is not possible to compare with realistic medical datasets as CLIP fails to generate meaningful concepts. Regardless, we have compared [18,33] with our method on CUB+OOD datasets in Table 1 of rebuttal pdf, and our model outperforms the accuracy of [18, 33], which is lower than the standard model.
> >The two losses (COOP loss Eq 3, and COL in Eq 6) seem to be proposed in prior works. The authors mentioned that COL is from [21].
>
> Orthogonality and alternative loss functions have been active research areas. [21] focused on establishing orthogonality in the embedding space between features of distinct classes. Therefore, the loss function was added just before the classifier layer. However, the COL we propose aims to induce orthogonality among the different concepts predicted. This introduces orthogonality between the respective linear layers associated with each concept. As a result, our formulation considerably deviates from that of [21], as we're introducing orthogonality within a distinct space and across different dimensions.
> >It feels like the COOP loss in Eq 3 [...] Do you mean that it is expected that [33] with joint training[..]
>
> We cannot make claims on how PCBM-h[33] would perform when jointly trained but it should be noted that the residual term in [33] is non-interpretable which might defeat the purpose of concept-based explanations while less accurate than the standard model. Hence unlike [33], we can achieve higher task accuracy while maintaining high concept accuracy (Table 2, main paper).
> >Large scale datasets results are missing (ImageNet).
>
> We have validated our method on 3 diverse datasets: TIL is a realistic medical dataset, AwA2 is a large-scale 13GB dataset and we have performed multiple realistic evaluations on distributional shifts. Most works on CBMs do not evaluate on ImageNet apart from [18] whose concept acquisition methodology is different.
> >For example, to describe f, g, h, it'll be useful to visualize the structure.
>
> We have added the figure in the rebuttal PDF.
>  >Also, since f and h actually differ in one linear layer, why do both of them still use theta? it seems phi is the parameter of g.
>
> $\theta$ is the encoder layer. It should be noted that f has multiple linear layers as well. Yes $\phi$ is the parameter of g. We shall make it clearer in the camera ready version.
> >It's not clear why the concepts are necessary need to be orthogonal.
>
> Briefly mentioned in Section 3, we introduce orthogonality to increase the separation between different concepts. Independent CBMs[12] which take binary (0/1) concept values as input into the concept predictor, exhibit the highest concept accuracy, suggesting enhanced interpretability. Simultaneously [8,16] indicated that information leakage via "soft" concept labels leads to low concept accuracy compromising explainability. COL introduces orthogonality constraints among different predicted concepts, promoting their independence and reducing the potential for information leakage between them. By encouraging orthogonality between concept representations, COL promotes learning of distinct, disentangled, and meaningful high-level concepts. COL leads to three significant applications: 1) enhances concept representation and accuracy (see Table 2 main paper on CUB), 2) improves task accuracy 3) helps to make more meaningful interventions i.e. with a lower intervention budget, one can achieve higher accuracy. Orthogonality in concept learning is crucial for models aiming to balance accuracy, explainability, and prevent information leakage.
> >Computation cost
>
> We have added this in Table 2 of the rebuttal PDF
>  >Please give more details on how Fig 1 is plotted.
>
> * L->R-The first two graphs show the importance of using human uncertainty(SCS) as a metric to select interventions as compared to concept uncertainty(CUS) and weightage(CWS). We show this in two ways:
> * First we plot for accuracy of the CBM model (as used by other baseline metrics [3,26]) when the annotator correctly intervenes on the concept. Intuitively, the accuracy should increase.
> * Second, consider that scenario is similar to first except the annotator has changed. The annotator with ill intentions changes the concept values to incorrect concepts (instead of GT concepts). Now we plot all of the metrics and intuitively the accuracy would reduce.
> * L->R Last two graphs of Figure 1 compare the random interventions of different baseline models. Since SCS was present for CUB only, we consider random interventions for TIL and CUB
> >In eq 5, how do you define intra-concepts? c_i= c_j clarification.
>
> In retrospect, we would like to rewrite as $d_1=\sum_{\substack{i,j\in B,\\\\c_i^a = c_j^a\\\\a\in A}}\frac{q_i^T q_j}{||q_i|| ||q_j||};d_2=\sum_{\substack{i,j\in B,\\\\ c_i^a\neq c_j^a \\\\a\in A}} \frac{q_i^T q_j}{||q_i||||q_j||}$ where A is total concepts. $c_i=c_j$ brings concept representation of the same label concept closer
> >Limitations
>
> In the original submission, it is part of Appendix B. We will move it to the main paper.
>
> We would like to thank the reviewer again for the valuable feedback. We ran the requested experiments and attempted to answer all of the concerns of the reviewer and would be happy to answer any further questions.

---

> > ### Comment · Reviewer_G8F4 · 2023-08-15
> > **Acknowledgement of rebuttal**
> >
> > Dear authors,
> >
> > Thank you for the detailed response and the additional clarification + experiments. The additional experiment comparison in rebuttal Table 1 with [18, 33] on the CUB did strengthen this paper with stronger evidence. Also, the clarification fig 1 makes it much clearer to understand the proposed Coop-CBM.
> >
> > I have no other concerns and will increase the rating to acceptance with rating 6. Please include the rebuttal results, including updated equation 5, Table 1-2, Fig 1 to the camera ready version to improve the quality of the paper.

---

> > > ### Author Response · Authors · 2023-08-15
> > > **Thank you for your response**
> > >
> > > We are grateful for the feedback provided by the reviewer, and we are glad that our reply addressed your concerns. Your feedback has strengthened our work. We will incorporate your feedback into the camera-ready version. We would be happy to address any further questions/suggestions that might come up until the end of the discussion period.

---

### Author Rebuttal · Authors · 2023-08-10

Global rebuttal.

---

### Author Response · Authors · 2023-08-21
**Thank you reviewers**

We thank all reviewers for their valuable time spent writing insightful reviews and engaging in fruitful discussions with us. Your feedback has significantly improved the manuscript.

---

### Decision · Program_Chairs · 2023-09-21

**Decision:**

Accept (poster)

**Comment:**

The reviewers and AC discussed this paper and read the authors’ response carefully. All the reviewers unanimously agreed to accept this paper. The authors are encouraged to address the concerns from the reviewers in the camera ready, such as

1.	Improve the discussion on related work

2.	Improve the writing of the paper

3.	Address the concern on the large number of hyperparameters.